# Learning General Causal Structures with Hidden Dynamic Process for Climate Analysis

## Abstract

The heart of climate analysis is a rational effort to understand the *causes* behind the *purely observational* data. Latent driving forces, such as atmospheric processes, play a critical role in temporal dynamics, and the task of inferring such latent forces is often a problem of Causal Representation Learning (CRL). Moreover, geographically nearby regions may directly interact with each other, and such direct causal relations among the observed data are often not modeled in traditional CRL, making the problem more challenging. In this paper, we propose a unified framework that can uncover not only the latent driving forces, but also the causal relations among the observed variables. We establish conditions under which the hidden dynamic process and the relations among the observed variables are simultaneously identifiable from time-series data. Even without parametric assumptions on the causal relations, we provide identifiability guarantees for recovering latent variables and the relations among the observed variables via contextual information. Guided by these insights, we propose a framework for nonparametric **Ca**usal **D**iscovery and **Re**presentation learning (**CaDRe**), based on a time-series generative model with structural constraints. Synthetic data validates our theoretical claims. On real-world climate datasets, CaDRe achieves competitive forecasting performance and offers the visualized causal graphs consistent with domain knowledge, which is expected to improve our understanding of the climate systems.

## 1 Introduction

Understanding the causal structure of climate systems is fundamental not only to scientific reasoning [59], but also to reliable modeling and prediction. Given the observed data with $d_x$ variables: $\mathbf{x}_t = [x_{t,1}, \ldots, x_{t,d_x}]$, our goal is twofold: (1) to discover the underlying latent variables $\mathbf{z}_t = [z_{t,1}, \ldots, z_{t,d_z}]$ and their temporal interactions, and (2) to identify causal relations among observed variables. To better understand this problem, we describe it using a causal modeling perspective. As depicted in Figure 1, latent drivers $\mathbf{z}_t$, such as pressure and precipitation [8], are not directly measured but significantly influence the observed dynamics. These latent processes evolve jointly and stochastically, exhibiting both *instantaneous* and *time-lagged* causal dependencies [43, 57]. They govern observable quantities $\mathbf{x}_t$ like temperature, which reflect underlying dynamics and also exhibit spatial interactions through emergent weather patterns, such as wind circulation systems.

Identifying these underlying hidden variables and temporal relations is the central objective of Causal Representation Learning (CRL) [61] problem. Recent advances in identifiability theory and practical algorithm design fall under the framework of nonlinear Independent Component Analysis (ICA). These approaches typically rely on auxiliary variables [23, 24, 22, 75], sparsity [32, 85, 86, 31, 5], or restricted generative functions [16], and generally assume a *noise-free* and *invertible* generation from $\mathbf{z}_t$ to $\mathbf{x}_t$, in order to *directly* recover latent space. However, climatic measurements exhibit

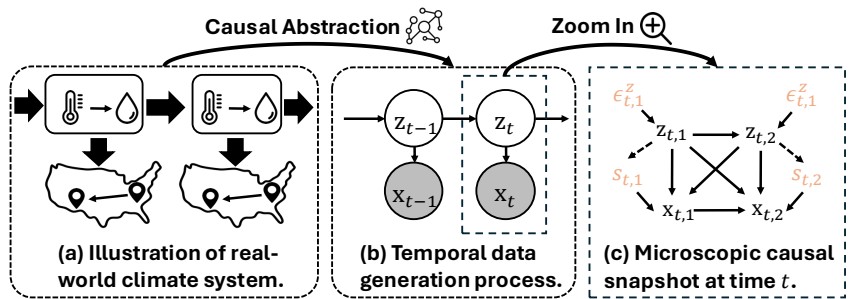

Figure 1: From climate system to causal graph. $\mathbf{x}_t$ represent observed data and $\mathbf{z}_t$ denotes unobserved variables behind $\mathbf{x}_t$, $\epsilon^z_t$ denotes the stochasticility in latent causal process, and $s_t$ denotes the noise variable varying with $\mathbf{z}_t$, *e.g.*, human activities [8].

both observational dependencies and stochastic noise, violating these assumptions and limiting the applicability of existing CRL approaches.

This problem can also be cast as the problem of causal discovery [65, 51] in the presence of latent processes. Causal discovery often relies on parametric models, such as linear non-Gaussianity [62], nonlinear additive [17, 30], post-nonlinear models [80], as well as nonparametric methods with [21, 55, 46] or without auxiliary variables [64, 84, 81]. However, generally speaking, they cannot identify latent variables, their interrelations, and their causal influence on observed variables. For example, Fast Causal Inference (FCI) algorithm [64] produces asymptotically correct results in the presence of latent confounders by exploiting conditional independence relations, but its result is often not informative enough; for instance, it cannot recover causally-related latent variables.

This above underscores the need for a unified framework capable of modeling both the observational causal structure, defined as the relations among the observed variables, and latent dynamic processes inherent to real-world climate systems. We understand the climate system through a causal lens and establish the identifiability guarantees for jointly recovering latent dynamics and observational causal graphs. Intuitively, the temporal structure enables leveraging contextual observable information to identify latent factors, while the inferred latent dynamics, in turn, modulate how observational causal graphs evolve. We instantiate this insight in a state-space Variational AutoEncoder (VAE), which can conduct nonparametric **Ca**usal **D**iscovery and **Re**presentation learning (**CaDRe**) simultaneously.

CaDRe employs parallel flow-based priors to *learn independent components* to reflect structural dependencies, and introduces gradient-based structural penalties on both latent transitions and decoders to ensure identifiability. Extensive synthetic experiments on the identification of latent representation learning and causal discovery validate our theoretical guarantees. On real-world climate data, CaDRe achieves competitive forecasting accuracy, indicating the effectiveness of the learned temporal process. The visualized causal graphs align with known scientific phenomena, *e.g.*, wind circulation and land–sea interactions, and further reveal structural patterns that may inspire new hypotheses in climate science.

## 2 Problem Setup

**Technical Notations.** We present the notations in a climate system, a terminology widely used in ICA literature [23]. We observed a time-series of observed variables $\mathbf{X} = [\mathbf{x}_1, \mathbf{x}_2, \cdots, \mathbf{x}_T]$, whereas their underlying factors $\mathbf{Z} = [\mathbf{z}_1, \mathbf{z}_2, \cdots, \mathbf{z}_T]$ are unobservable. Regarding the system in one time-step, as depicted in Figure 1, it consists of observed variables $\mathbf{x}_t := [x_{t,i}]_{i \in \mathcal{I}}$ with index set $\mathcal{I} = \{1, 2, \ldots, d_x\}$, and latent variables $\mathbf{z}_t := [z_{t,j}]_{j \in \mathcal{J}}$ indexed by $\mathcal{J} = \{1, 2, \ldots, d_z\}$. Let $\mathbf{pa}(\cdot)$ denotes the parent variables, $\mathbf{pa}_O(\cdot)$ refers to observable parents, and $\mathbf{pa}_L(\cdot)$ indicates the latent parents. In particular, $\mathbf{pa}_L(\cdot)$ comprises latent variables from both the current and previous time step. Throughout the paper, the hat notation, *e.g.*, $\hat{\mathbf{x}}_t$, denotes estimated variables or functions.

**Data Generating Process.** Suppose we have observed the time series data with discrete timestamps $\mathbf{X} = [\mathbf{x}_1, \mathbf{x}_2, \ldots, \mathbf{x}_T]$. We translate how a climate system evolves to the following Structural

Equation Model (SEM) [51] at each discrete time step:

$$x_{t,i} = \underbrace{g_i(\mathbf{pa}_O(x_{t,i}), \mathbf{pa}_L(x_{t,i}), s_{t,i})}_{\text{effects from } \mathbf{x}_t \text{ and } \mathbf{z}_t}, \quad z_{t,j} = \underbrace{f_j(\mathbf{pa}_L(z_{t,j}), \epsilon_{t,j}^z)}_{\text{effects from } \mathbf{z}_{t-1} \text{ and } \mathbf{z}_t}, \quad s_{t,i} = \underbrace{g_{s_i}(\mathbf{z}_t, \epsilon_{t,i}^x)}_{\text{noise conditioned on } \mathbf{z}_t}, \quad (1)$$

where $g_i$ and $f_j$ are differentiable functions, and noise terms $\epsilon_{t,j}^z \sim p_{\epsilon_{z_j}}, \epsilon_{t,i}^x \sim p_{\epsilon_{x_i}}$ are mutually independent for $\mathcal{I}$ and $\mathcal{J}$. As discussed in the introduction, the observed variable $x_{t,i}$ may be influenced by other observed components $\mathbf{x}_{t,\backslash i}$ and the latent variables $\mathbf{z}_t$. For example, temperature in a specific region may be governed by latent drivers such as solar radiation, and also be affected by neighboring regions through heat transfer. The stochastic term $s_{t,i}$, depending on latent variables $\mathbf{z}_t$, is designed to capture inherent climatic variability, such as perturbations introduced by human activities on $CO_2$ [66]. The latent variable $z_{t,j}$ evolves according to both instantaneous interactions with other components $\mathbf{z}_{t,\backslash j}$ and time-lagged dependencies from the previous step $\mathbf{z}_{t-1}$. Aiming at reliably discovering causal graphs, we additionally adopt an assumption [65] in causal discovery:

**Assumption 1.** *The distribution over* $(\mathbf{X}, \mathbf{Z})$ *is Markov and faithful to a Directed Acyclic Graph (DAG).*

Based on the generation process, we formally define the identifiability of latent space, latent variables, and observational causal structure, each serving as a prerequisite for reliable climate analysis.

**Definition 1** (Identifiability Criteria). *Let* $\mathbf{X} = \{\mathbf{x}_1, \ldots, \mathbf{x}_T\}$ *be a sequence of observed variables generated by the true latent causal processes specified by* $(f, g, g_s, p(\epsilon_t^z), p(\epsilon_t^x))$ *as described in Equation 1. A learned generative model* $(\hat{f}, \hat{g}, \hat{g}_s, \hat{p}(\epsilon_t^z), \hat{p}(\epsilon_t^x))$ *is said to be* observationally equivalent *to the true model if it induces the same distribution over observations, i.e.,* $p_{\hat{f}, \hat{g}, \hat{g}_s}(\hat{\mathbf{X}}) = p_{f, g, g_s}(\mathbf{X})$. *Under this equivalence, we define the following identifiability properties:*

  i. *(Identifiability of Latent Space):* $\hat{\mathbf{z}}_t = h(\mathbf{z}_t)$ *for all t, where* $h : \mathbb{R}^{d_z} \to \mathbb{R}^{d_z}$ *is invertible.*

  ii. *(Component-Wise Identifiability of Latent Variables):* $\hat{z}_{t,i} = h_i(z_{t,\pi(i)})$ *for all t and i, where* $\pi$ *is a permutation over* $\{1, \ldots, d_z\}$ *and* $h_i : \mathbb{R} \to \mathbb{R}$ *are invertible functions.*

  iii. *(Identifiability of Observational Causal Graph): The estimated causal parents of each observed variable match the ground truth, i.e.,* $\hat{\mathbf{pa}}_O(x_{t,i}) = \mathbf{pa}_O(x_{t,i})$ *for all t and i.*

# 3 Identification Theory

Given the above definitions and goals, we first establish the identifiability of the latent space in Theorem 1, and further show that the latent causal process is identifiable under a sparse latent process [31, 35] in Theorem A.3. We then draw a formal connection between the SEM and nonlinear ICA with latent variables, which are shown to describe the same data generating processes in Theorem 2. This connection nourishes a *functional equivalence* for computing the causal graphs through the mixing structure of ICA in Theorem 2. Finally, we prove the identifiability of the nonlinear ICA with latent variables in Theorem 3 by leveraging the cross-derivative condition [36], subsequently identifying observational causal graphs in SEM via functional equivalence.

## 3.1 Latent Space Recovery and Latent Variables Identification

In this section, we aim to characterize the relationships between ground truth $\mathbf{z}_t$ and estimated $\hat{\mathbf{z}}_t$. We consider, without loss of generality, a first-order Markov structure, in which three consecutive observations $\{\mathbf{x}_{t-1}, \mathbf{x}_t, \mathbf{x}_{t+1}\}$ are used as contextual information. The generalization to higher-order Markov structures is discussed in Appendix D.1. To formalize the stochastic generation process, we introduce an operator $L$ [12] to represent distribution-level transformations, that is, how one probability distribution is pushed forward to another. Given two random variables $a$ and $b$ with supports $\mathcal{A}$ and $\mathcal{B}$ respectively, the transformation $p_a \mapsto p_b$ is formalized as:

$$p_b = L_{b|a} \circ p_a, \text{ where } L_{b|a} \circ p_a := \int_{\mathcal{A}} p_{b|a}(\cdot \mid a) p_a(a) da. \quad (2)$$

For example, operators $L_{\mathbf{x}_{t+1}|\mathbf{z}_t}$ and $L_{\mathbf{x}_{t-1}|\mathbf{x}_{t+1}}$ represent the distributional transformations $p_{\mathbf{z}_t} \mapsto p_{\mathbf{x}_{t+1}}$ and $p_{\mathbf{x}_{t+1}} \mapsto p_{\mathbf{x}_{t-1}}$, respectively. With the help of this operator, we turn to addressing the problem of "causally-related" observed variables in recovering the latent space. When the observed causal graph forms a DAG, information is preserved along the causal pathways without getting trapped into a *self-loop*. This suggests that causal influence can be traced back to its source by following the reverse direction of the DAG through the "short reaction lag" [14]. Formally, this

121 implies that the associated operator in the generative process must be *injective*, ensuring that the
122 transformation retains full distributional information. We formalize this as follows:

123 **Lemma 1.** *(Injective DAG Operator) Under Assumption 1, $L_{\mathbf{x}_t|\mathbf{s}_t}$ is injective for all $t \in \mathcal{T}$.*

124 This result implies that the nonlinear causal DAG over $\mathbf{x}_t$ does not disturb the recovery of the latent
125 space. Building on this, we now address a more fundamental challenge: the latent variable $\mathbf{z}_t$ cannot
126 be recovered from a single noisy observation $\mathbf{x}_t$, as the stochasticity makes the value-level mapping
127 ill-posed. Instead, we seek identifiability at the distributional level. Notably, adjacent observations
128 $\mathbf{x}_{t-1}$ and $\mathbf{x}_{t+1}$ contain non-trivial information about $\mathbf{z}_t$ if they exhibit *minimal changes*. We propose
129 the following theorem to provide *nonparametric* identifiability of the latent submanifold based on
130 distributional variations captured by contextual measurements.

131 **Theorem 1.** *(Identifiability of Latent Space) Suppose observed variables and hidden variables*
132 *follow the data-generating process in Eq. (1), and estimated observations match the true joint*
133 *distribution of $\{\mathbf{x}_{t-1}, \mathbf{x}_t, \mathbf{x}_{t+1}\}$ as illustrated in Definition 1. The following assumptions are imposed:*

134 *A1 (Computable Probability:) The joint, marginal, and conditional distributions of $(\mathbf{x}_t, \mathbf{z}_t)$ are all*
135 *bounded and continuous.*

136 *A2 (Contextual Variability:) The operators $L_{\mathbf{x}_{t+1}|\mathbf{z}_t}$ and $L_{\mathbf{x}_{t-1}|\mathbf{x}_{t+1}}$ are injective and bounded.*

137 *A3 (Latent Drift:) For any $\mathbf{z}_t^{(1)}, \mathbf{z}_t^{(2)} \in \mathcal{Z}_t$ where $\mathbf{z}_t^{(1)} \neq \mathbf{z}_t^{(2)}$, we have $p(\mathbf{x}_t|\mathbf{z}_t^{(1)}) \neq p(\mathbf{x}_t|\mathbf{z}_t^{(2)})$.*

138 *A4 (Differentiability:) There exists a functional $M$ such that $M\left[p_{\mathbf{x}_t|\mathbf{z}_t}(\cdot \mid \mathbf{z}_t)\right] = h_z(\mathbf{z}_t)$ for all*
139 *$\mathbf{z}_t \in \mathcal{Z}_t$, where $h$ is differentiable.*

140 *Then we have $\hat{\mathbf{z}}_t = h_z(\mathbf{z}_t)$, where $h_z : \mathbb{R}^{d_z} \to \mathbb{R}^{d_z}$ is an invertible and differtiable function.*

141 **Discussion on Assumptions.** As presented, A1 is a moderate condition for computable density
142 fucntions. A2 introduces sufficient distributional variability, formalized via injectivity at the density
143 level. A3 ensures that distinct values of $\mathbf{z}_t$ induce distinct conditionals $p(\mathbf{x}_t \mid \mathbf{z}_t)$, which is violated
144 only when two values of $\mathbf{z}_t$ yield identical distributions. A4 requires that the mapping from $\mathbf{z}_t$ to
145 $p(\mathbf{x}_t \mid \mathbf{z}_t)$ is differentiable—a condition naturally satisfied by models based on differentiable neural
146 networks, such as VAEs. Please refer to Appendix A.2 for detailed discussions.

147 **Proof Sketch and Contributions.** The complete proof is deferred to Appendix A. Prior work on
148 nonparametric identifiability [18, 6] relies on partially knowing the function form of generation
149 mechanism $g$, and yields only distribution-level identifiability, *i.e.*, $p_{\hat{\mathbf{z}}_t} = p_{\mathbf{z}_t}$. In contrast, our
150 approach requires no such prior knowledge and achieves identifiability at the value level, a more
151 informative result. As depicted in Eq. (A19), we begin by proving the uniqueness of the posterior
152 collection $\{p(\mathbf{x}_t \mid \hat{\mathbf{z}}_t)\}_{\hat{\mathcal{Z}}_t}$, where the unordered set unveils the existence of a relabeling function $h$
153 on the conditioning variables. A3 then ensures a one-to-one correspondence between $\mathbf{z}_t$ and the
154 posteriors $p(\mathbf{x}_t \mid \hat{\mathbf{z}}_t)$, thereby ruling out degenerate mappings from posteriors to values.

$$\boxed{\{p_{\mathbf{x}_t|\mathbf{z}_t}(\cdot \mid \mathbf{z}_t)\}_{\mathcal{Z}_t} = \{p_{\mathbf{x}_t|\hat{\mathbf{z}}_t}(\cdot \mid \hat{\mathbf{z}}_t)\}_{\hat{\mathcal{Z}}_t}} \Rightarrow \boxed{p_{\mathbf{x}_t|\mathbf{z}_t}(\mathbf{x}_t \mid h_z(\mathbf{z}_t)) = p_{\mathbf{x}_t|\hat{\mathbf{z}}_t}(\mathbf{x}_t \mid \hat{\mathbf{z}}_t)} \Rightarrow \boxed{\hat{\mathbf{z}}_t = h_z(\mathbf{z}_t)}$$

155 Finally, A4 pins down the $h$ to be differentiable. After recovering the latent space, we aim to enhance
156 interpretability by ensuring that each latent component corresponds to a distinct physical variable.
157 To achieve this, we introduce a sparsity assumption on the latent dynamics, which is motivated by
158 that physical climate factors—such as solar radiation, atmospheric pressure, or ocean currents—tend
159 to exhibit localized sparse influences. Please refer to Appendix A.3 for the theorem regarding
160 *component-wise identifiability of latent variables.*

### 3.2 Nonparametric Causal Discovery with the Hidden Dynamic Process

162 Building upon the results on recovering latent representations, we now seek to identify general
163 nonlinear causal graphs over $\mathbf{x}_t$, even if they are modulated by a hidden dynamic process. Recent
164 works [46, 55] extend the ICA-based Causality Discovery (CD) [62] to nonparametric settings via
165 nonlinear ICA [25]. However, these methods are not applicable in the presence of latent confounders.
166 To overcome this limitation, we establish a refined connection between SEMs and nonlinear ICA.

167 **Lemma 2.** *(Nonlinear SEM $\Leftrightarrow$ Nonlinear ICA) There exists a function $m_i$, which is differentiable*
168 *w.r.t. $s_{t,i}$ and $\mathbf{x}_t$, for any fixed $s_{t,i}$ and $\mathbf{z}_t$, such that the following two representations,*

$$x_{t,i} = g_i(\mathbf{pa}_O(x_{t,i}), \mathbf{pa}_L(x_{t,i}), s_{t,i}) \quad and \quad x_{t,i} = m_i(\mathbf{z}_t, \mathbf{s}_t) \tag{3}$$

169 *describe the same data-generating process. That is, both expressions yield the same value of $x_{t,i}$.*

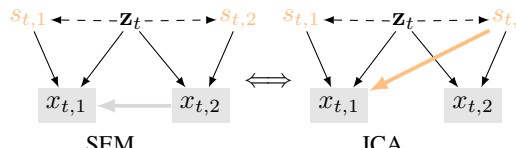

Figure 2: **Equivalent SEM and ICA.** The gray line in SEM denotes the influence $x_{t,2} \rightarrow x_{t,1}$ through the observation causal relation, which is equivalently represented as an indirect effect (the orange line): $s_{t,2} \dashrightarrow x_{t,1}$ in ICA, which can be decomposed into $s_{t,2} \rightarrow x_{t,2}$ and $x_{t,2} \rightarrow x_{t,1}$.

After establishing this equivalence, we proceed to perform CD via the nonlinear ICA with latent variables. We begin by introducing the Jacobian matrices on this data generating process, as they serve as proxies for the (nonlinear) adjacency matrix. For all $(i,j) \in \mathcal{I} \times \mathcal{I}$, we define $[\mathbf{J}_m(\mathbf{s}_t)]_{i,j} = \frac{\partial x_{t,i}}{\partial s_{t,j}}$, $[\mathbf{J}_g(\mathbf{x}_t)]_{i,j} = \frac{\partial x_{t,i}}{\partial x_{t,j}}$, and $\mathbf{D}_m(\mathbf{s}_t) = \mathrm{diag}(\frac{\partial x_{t,1}}{\partial s_{t,1}}, \frac{\partial x_{t,2}}{\partial s_{t,2}}, \dots, \frac{\partial x_{t,d_x}}{\partial s_{t,d_x}})$, $\mathbf{I}_{d_x}$ is the identity matrix in $\mathbb{R}^{d_x \times d_x}$. Here, $\mathbf{J}_m(\mathbf{s}_t)$ corresponds to the mixing process of nonlinear ICA, as described on the R.H.S. of Eq. (3). Note that $\mathbf{J}_g(\mathbf{x}_t)$ signifies the observational causal graph in the nonlinear SEM, the L.H.S. of Eq. (3), provided the faithfulness assumption outlined below holds.

**Assumption 2** (Functional Faithfulness). *The causal adjacency structure among observed variables is given by the support of the Jacobian matrix* $\mathbf{J}_g(\mathbf{x}_t)$.

This assumption implies *edge minimality* in causal graphs, analogous to the structural minimality discussed in [52] (Remark 6.6) and minimality in [79], which enables us to establish a equivalence between the observational causal graph in SEM and the mixing structure in nonlinear ICA.

**Theorem 2.** *(Functional Equivalence)* *Consider the two types of data generating process described in Eq. (3), the following equation always holds:*

$$\mathbf{J}_g(\mathbf{x}_t)\mathbf{J}_m(\mathbf{s}_t) = \mathbf{J}_m(\mathbf{s}_t) - \mathbf{D}_m(\mathbf{s}_t). \tag{4}$$

**Proof Sketch.** Following the depiction of the SEM, the flow of information can be traced starting from the observed variables $\mathbf{x}_t$. The DAG structure ensures that the sources are the latent variables and the independent noise, implying that the data generation process conforms to a specific nonlinear ICA: $[\mathbf{z}_t, \boldsymbol{\epsilon}_{x_t}] \Rightarrow \mathbf{x}_t$, where $\mathbf{z}_t$ is characterized as a conditional prior. Refer to Appendix A.4 for a detailed proof. We establish two results that strengthen the SEM–ICA connection by relaxing modeling assumptions and enabling its practical application within generative models.

**Corollary 2.1.** *Under Assumption 1, given any* $\mathbf{z}_t \in \mathcal{Z}_t$, $\mathbf{J}_m(\mathbf{s}_t)$ *is a invertible matrix.*

This result unveils that the DAG structure among observed variables implies the invertibility of the mixing function $m$ in the nonlinear ICA. As a direct consequence, by left-multiplying both sides of Eq. (4) with $\mathbf{J}_m^{-1}(\mathbf{s}_t)$, we obtain the following expression:

**Corollary 2.2.** *Observational causal graphs are represented by* $\mathbf{J}_g(\mathbf{x}_t) = \mathbf{I}_{d_x} - \mathbf{D}_m(\mathbf{s}_t)\mathbf{J}_m^{-1}(\mathbf{s}_t)$.

Building upon these SEM–ICA connections, we derive sufficient conditions under which the observational causal graph becomes identifiable in virtue of the recovered latent processes.

**Theorem 3.** *(Identifiability of Observational Causal Graph)* *Let* $\mathbf{A}_{t,k} = \log p(\mathbf{s}_{t,k}|\mathbf{z}_t)$, *assume that* $\mathbf{A}_{t,k}$ *is twice differentiable in* $s_{t,k}$ *and is differentiable in* $z_{t,l}$, *where* $l = 1, 2, ..., d_z$. *Suppose Assumption 1, 2 holds true, and*

*A5 (Generation Variability). For any estimated* $\hat{g}_m$ *that makes* $\mathbf{x}_t = \hat{\mathbf{x}}_t = \hat{m}(\hat{\mathbf{z}}_t, \hat{\mathbf{s}}_t)$, *let*

$$\mathbf{V}(t,k) := \left[ \frac{\partial^2 \mathbf{A}_{t,k}}{\partial s_{t,k} \partial z_{t,1}}, \dots, \frac{\partial^2 \mathbf{A}_{t,k}}{\partial s_{t,k} \partial z_{t,d_z}} \right], \mathbf{U}(t,k) := \left[ \frac{\partial^3 \mathbf{A}_{t,k}}{\partial s_{t,k} \partial^2 z_{t,1}}, \dots, \frac{\partial^3 \mathbf{A}_{t,k}}{\partial s_{t,k} \partial^2 z_{t,d_z}} \right]^T,$$

*where for* $k = 1, 2, \dots, d_x$, $2d_x$ *vector functions* $\mathbf{V}(t,1), \dots, \mathbf{V}(t, d_x), \mathbf{U}(t,1), \dots, \mathbf{U}(t, d_x)$ *are linearly independent. Then we attain ordered component-wise identifiability (Definition 5), and the structure of the observational causal graph is identifiable, i.e.,* $supp(\mathbf{J}_g(\mathbf{x}_t)) = supp(\mathbf{J}_{\hat{g}}(\hat{\mathbf{x}}_t))$.

**Proof Sketch.** The core idea of the proof extends the notion of component-wise identifiability. Recall from data generating process 1 that satisfies $s_{t,i} \perp\!\!\!\perp s_{t,j} \mid \mathbf{z}_t$ for all $i \neq j$. From Theorem 1, we know that block-level information about $\mathbf{z}_t$ is identifiable and can be treated as a continuous conditioning domain [23]. To eliminate the permutation ambiguity, we further exploit the structural constraints encoded by the DAG over observed variables. The full proof is provided in Appendix A.7.

Table 1: Attributes of causal discovery that can apply to time-series. A check denotes that a method has an attribute or result, whereas a cross denotes the opposite.

| Method | Nonparametric | Latent Variables | Latent Causal Graph | Observational Causal Graph | No Equivalence Classes |
|---|---|---|---|---|---|
| NESSM [20] | ✗ | ✗ | ✗ | ✓ | ✓ |
| CD-NOD [21] | ✓ | ✗ | ✗ | ✓ | ✗ |
| FCI [64] | ✓ | ✓ | ✗ | ✓ | ✗ |
| LPCMCI [15] | ✓ | ✓ | ✗ | ✓ | ✗ |
| CDSD [5] | ✓ | ✓ | ✓ | ✗ | ✓ |
| CaDRe | ✓ | ✓ | ✓ | ✓ | ✓ |

In summary, these results establish a clear pipeline for reliably learning observational causal graphs through latent variable identification, noise component identification, and structure identification.

$$\boxed{\hat{\mathbf{z}}_t = h_z(\mathbf{z}_t)} \Rightarrow \boxed{\hat{s}_{t,i} = h_s(s_{t,\pi(i)})} \Rightarrow \boxed{\mathrm{supp}(\mathbf{J}_{\hat{m}}) = \mathrm{supp}(\mathbf{J}_m)} \Rightarrow \boxed{\mathrm{supp}(\mathbf{J}_{\hat{g}}) = \mathrm{supp}(\mathbf{J}_g)}$$

**Method Comparison.** As summarized in Table 1, CaDRe supports versatile causal discovery across multiple settings while addressing these challenges in a unified framework. NESSM [20] models time-varying causal strengths but assumes causal sufficiency and restricts the causal model to a linear form, making it a special case of CaDRe. CD-NOD [21] needs nonstationarity for causal discovery and does not model latent variables, and suffers from equivalence classes. FCI [64] requires no auxiliary assumptions but cannot recover latent variables and is limited to Partial Ancestral Graphs (PAGs). LPCMCI [15] considers both latent variables and observational structure but does not model latent temporal dynamics. CDSD [5] is designed for climate settings and assumes sparse causal mechanisms, yet lacks a modeling framework for observational interactions in climate systems.

## 4  Estimation Methodology

Our theoretical insights shed light on the practical implementations. As shown in Figure 4, we instantiate these insights into an estimation framework for **Ca**usal **D**iscovery and causal **Re**presentation learning (**CaDRe**) in the nonparametric setting, enabling direct inference of causal structures.

**Overall Architecture.**  The proposed architecture is built upon the variational autoencoder [27]. In light of data generating process 1, we establish the Evidence Lower BOund (ELBO) as follows:

$$
\begin{aligned}
\mathcal{L}_{ELBO} = \mathbb{E}_{q(\mathbf{s}_{1:T}|\mathbf{x}_{1:T})} & \left[\log p(\mathbf{x}_{1:T} \mid \mathbf{s}_{1:T}, \mathbf{z}_{1:T})\right] - \\
& \lambda_1 D_{\mathrm{KL}}\left(q(\mathbf{s}_{1:T} \mid \mathbf{x}_{1:T}) \,\|\, p(\mathbf{s}_{1:T} \mid \mathbf{z}_{1:T})\right) - \lambda_2 D_{\mathrm{KL}}\left(q(\mathbf{z}_{1:T} \mid \mathbf{x}_{1:T}) \,\|\, p(\mathbf{z}_{1:T})\right),
\end{aligned}
\tag{5}
$$

where $\lambda_1$ and $\lambda_2$ are hyperparameters, and $D_{KL}$ represents the Kullback-Leibler divergence. We set $\lambda_1 = 4 \times 10^{-3}$ and $\lambda_2 = 1.0 \times 10^{-2}$ to achieve the best performance. In Figure 4, the `z-encoder`, `s-encoder` and `decoder` are implemented by Multi-Layer Perceptrons (MLPs) as follows:

$$\mathbf{z}_{1:T} = \phi(\mathbf{x}_{1:T}), \quad \mathbf{s}_{1:T} = \eta(\mathbf{x}_{1:T}), \quad \hat{\mathbf{x}}_{1:T} = \psi(\mathbf{z}_{1:T}, \mathbf{s}_{1:T}), \tag{6}$$

respectively, where the `z-encoder` $\phi$ learns the latent variables through denoising, and `s-encoder` $\psi$ and `decoder` $\eta$ approximate functions for encoding $\mathbf{s}_t$ and reconstructing observations, respectively.
**Prior Estimation of $\mathbf{z}_t$ and $\mathbf{s}_t$.** We propose using the `s-prior` network and `z-prior` network to recover the independent noise $\hat{\boldsymbol{\epsilon}}_t^x$ and $\hat{\boldsymbol{\epsilon}}_t^z$, respectively, thereby estimating the prior distribution of latent variables $\hat{\mathbf{z}}_t$ and dependent noise $\hat{\mathbf{s}}_t$. Specifically, we first let $r_i$ be the $i$-th learned inverse transition function that take the estimated latent variables as input to recover the noise term, e.g., $\hat{\epsilon}_{t,i}^z = r_i(\hat{\mathbf{z}}_{t-1}, \hat{\mathbf{z}}_t)$. Each $r_i$ is implemented by MLPs. Sequentially, we devise a transformation $\kappa := \{\hat{\mathbf{z}}_{t-1}, \hat{\mathbf{z}}_t\} \to \{\hat{\mathbf{z}}_{t-1}, \hat{\boldsymbol{\epsilon}}_t^z\}$, whose Jacobian can be formalized as $\mathbf{J}_\kappa = \begin{pmatrix} \mathbf{I} & 0 \\ \mathbf{J}_d(\hat{\mathbf{z}}_{t-1}) & \mathbf{J}_r(\hat{\mathbf{z}}_t) \end{pmatrix}$.
Then we have  Eq. (7) derived from normalizing flow to estimate the prior distribution:

$$\log p(\hat{\mathbf{z}}_t, \hat{\mathbf{z}}_{t-1}) = \log p(\hat{\mathbf{z}}_{t-1}, \hat{\boldsymbol{\epsilon}}_t^z) + \log |\frac{\partial r_i}{\partial \hat{z}_{t,i}}|. \tag{7}$$

According to the generation process, the noise $\epsilon_{t,i}^z$ is independent of $\mathbf{z}_{t-1}$, allowing us to enforce independence on the estimated noise term $\hat{\epsilon}_{t,i}^z$ with $\mathcal{D}_{KL}$. Consequently, Eq. (7) can be rewritten as:

$$\log p(\hat{\mathbf{z}}_{1:T}) = p(\hat{\mathbf{z}}_1) \prod_{\tau=2}^{T} \left( \sum_{i=1}^{d_z} \log p(\hat{\epsilon}_{\tau,i}^z) + \sum_{i=1}^{d_z} \log |\frac{\partial r_i}{\partial \hat{z}_{\tau,i}}| \right), \tag{8}$$

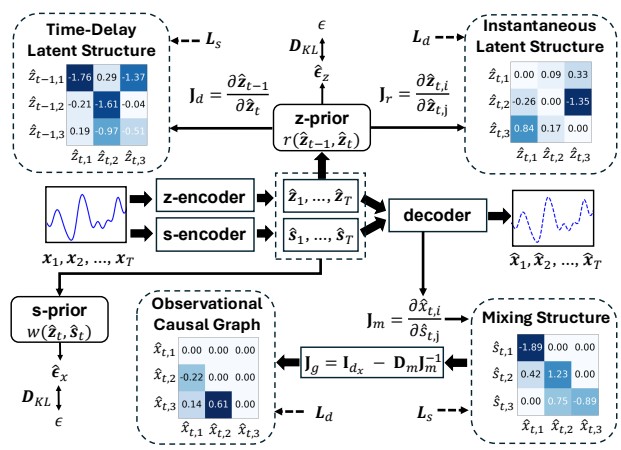

Figure 3: **The estimation procedure of CaDRe.** The model framework includes two encoders: `z-encoder` for extracting latent variables $\mathbf{z}_t$, and `s-encoder` for extracting $\mathbf{s}_t$. A `decoder` reconstructs observations from these variables. Additionally, prior networks estimate the prior distribution using normalizing flow, target on learning causal structure based on the Jacobian matrix. $\mathcal{L}s$ imposes a sparsity constraint and $\mathcal{L}d$ enforces the DAG structure on Jacobian matrix. $D_{KL}$ enforces an independence constraint on the estimated noise by minimizing its KL divergence w.r.t. $\mathcal{N}(0, \mathbf{I})$. In summary, this method learns independent noise to inversely infer the causal structures.

Table 2: **Results under Varying Observational Dimensionality** ($d_x$). Each setting is repeated with 5 random seeds. For evaluation, the best-converged result per seed is selected to avoid local minima.

| $d_z$ | $d_x$ | SHD ($\mathbf{J}_{\hat{g}}(\hat{\mathbf{x}}_t)$) | TPR | Precision | MCC ($\mathbf{s}_t$) | MCC ($\mathbf{z}_t$) | SHD ($\mathbf{J}_r(\hat{\mathbf{z}}_t)$) | SHD ($\mathbf{J}_d(\hat{\mathbf{z}}_{t-1})$) | $R^2$ |
|---|---|---|---|---|---|---|---|---|---|
| | 3 | 0 | 1 | 1 | $0.9775_{\pm 0.01}$ | $0.9721_{\pm 0.01}$ | $0.27_{\pm 0.05}$ | $0.26_{\pm 0.03}$ | $0.90_{\pm 0.05}$ |
| | 6 | $0.18_{\pm 0.06}$ | $0.83_{\pm 0.03}$ | $0.80_{\pm 0.04}$ | $0.9583_{\pm 0.02}$ | $0.9505_{\pm 0.01}$ | $0.24_{\pm 0.06}$ | $0.33_{\pm 0.09}$ | $0.92_{\pm 0.01}$ |
| 3 | 8 | $0.29_{\pm 0.05}$ | $0.78_{\pm 0.05}$ | $0.76_{\pm 0.04}$ | $0.9020_{\pm 0.03}$ | $0.9601_{\pm 0.03}$ | $0.36_{\pm 0.11}$ | $0.31_{\pm 0.12}$ | $0.93_{\pm 0.02}$ |
| | 10 | $0.43_{\pm 0.05}$ | $0.65_{\pm 0.08}$ | $0.63_{\pm 0.14}$ | $0.8504_{\pm 0.07}$ | $0.9652_{\pm 0.02}$ | $0.29_{\pm 0.04}$ | $0.40_{\pm 0.05}$ | $0.92_{\pm 0.02}$ |
| | 100* | $0.17_{\pm 0.02}$ | $0.80_{\pm 0.05}$ | $0.81_{\pm 0.02}$ | $0.9131_{\pm 0.02}$ | $0.9565_{\pm 0.02}$ | $0.21_{\pm 0.01}$ | $0.29_{\pm 0.10}$ | $0.93_{\pm 0.03}$ |

where $p(\hat{\epsilon}^z_{\tau,i})$ is assumed to follow a Gaussian distribution. Similarly, we estimate the prior of $\mathbf{s}_t$ using $\hat{\epsilon}^x_{t,i} = w_i(\hat{\mathbf{z}}_t, \hat{\mathbf{s}}_t)$, and model the transformation between $\hat{\mathbf{s}}_t$ and $\hat{\mathbf{z}}_t$ as follows:

$$\log p\left(\hat{\mathbf{s}}_{1:T} \mid \hat{\mathbf{z}}_{1:T}\right) = \prod_{\tau=1}^{T} \left( \sum_{i=1}^{d_x} \log p\left(\hat{\epsilon}^x_{\tau,i}\right) + \sum_{i=1}^{d_x} \log \left| \frac{\partial w_i}{\partial \hat{s}_{\tau,i}} \right| \right). \tag{9}$$

Specifically, to ensure the conditional independence of $\hat{\mathbf{z}}_t$ and $\hat{\mathbf{s}}_t$, we using $\mathcal{D}_{KL}$ to minimize the KL divergence from the distributions of $\hat{\epsilon}^x_t$ and $\hat{\epsilon}^z_t$ to the distribution $\mathcal{N}(0, \mathbf{I})$.

**Structure Learning.** The variables $r_i$ and $w_i$ are designed to capture causal dependencies among latent and observed variables, respectively. We denote $\mathbf{J}_d(\hat{\mathbf{z}}_{t-1})$ as the Jacobian matrix of the function $r$, which implies the estimated time-lagged latent causal structure; $\mathbf{J}_r(\hat{\mathbf{z}}_t)$, which implies the estimation of instantaneous latent causal structure; and $\mathbf{J}_{\hat{g}}(\hat{\mathbf{x}}_t)$, which implies the estimated observational causal graph. Considering the observational causal graph, we compute $\mathbf{J}_{\hat{g}_m}(\hat{\mathbf{s}}_t)$ from the `decoder`, and instantly obtain the observational causal graph $\mathbf{J}_{\hat{g}}(\hat{\mathbf{x}}_t)$ via Corollary 2.2. Notably, the entries of $\mathbf{J}_{\hat{g}}(\hat{\mathbf{x}}_t)$ vary with other variables such as $\hat{\mathbf{z}}_t$, resulting in a DAG that could change over time. For the latent structure, we directly compute $\mathbf{J}_d(\hat{\mathbf{z}}_{t-1})$ and $\mathbf{J}_r(\hat{\mathbf{z}}_t)$ from `z-prior` network as the time-lagged structure and instantaneous structure in latent space, respectively. To prevent redundant edges and cycles, a sparsity penalty $\mathcal{L}_s$ are imposed on each learned structure, and DAG constraints $\mathcal{L}_d$ are imposed on the observational causal graph and instantaneous latent causal DAG. Specifically, the Markov network structure for latent variables is derived as $\mathcal{M}(\mathbf{J}) = (\mathbf{I} + \mathbf{J})^\top (\mathbf{I} + \mathbf{J}) - \mathbf{I}$. Formally, we define these penalties as follows:

$$\sum \mathcal{L}_s = \|\mathcal{M}(\mathbf{J}_r(\hat{\mathbf{z}}_t))\|_1 + \|\mathcal{M}(\mathbf{J}_d(\hat{\mathbf{z}}_{t-1}))\|_1 + \|\mathbf{J}_{\hat{g}}(\hat{\mathbf{x}}_t)\|_1, \quad \sum \mathcal{L}_d = \mathcal{D}(\mathbf{J}_{\hat{g}}(\hat{\mathbf{x}}_t)) + \mathcal{D}(\mathbf{J}_r(\hat{\mathbf{z}}_t)). \tag{10}$$

where $\mathcal{D}(A) = \text{tr}\left[(I + \frac{1}{m}A \circ A)^m\right] - m$ is the DAG constraint from [77], with $A$ being an $m$-dimensional matrix. $\|\cdot\|_1$ denotes the matrix $l_1$ norm. In summary, the overall loss function of the CaDRe model integrates ELBO and penalties for structural constraints, which is formalized as:

$$\mathcal{L}_{ALL} = \mathcal{L}_{ELBO} + \alpha \sum \mathcal{L}_s + \beta \sum \mathcal{L}_d, \tag{11}$$

where $\alpha = 1.0 \times 10^{-4}$ and $\beta = 5.0 \times 10^{-5}$ are hyperparameters. The discussions about hyperparameter selections and their effects on performance are given in Appendix C.1.

Figure 5: **Comparison with Constraint-Based CD.** We set $d_x = 6$ and $d_z = 3$. We run experiments using 5 different random seeds, and report the average performance on evaluation metrics.

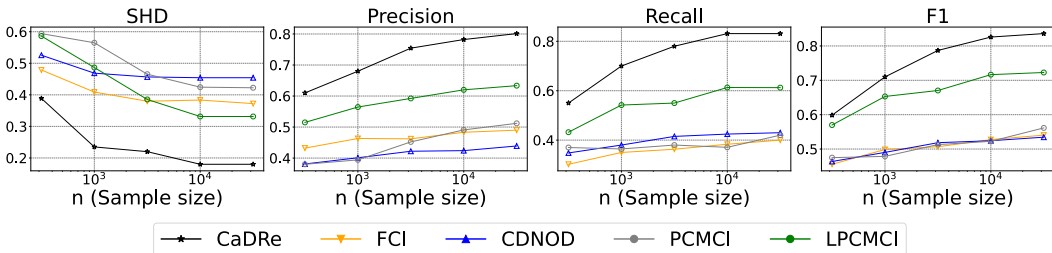

Table 3: **Identification Results on Simulated Data.** We set the dimensions as $d_z = 3$ and $d_x = 10$, and consider three scenarios according to our theory: *i) Independent*: $z_{t,i}$ and $z_{t,j}$ are conditionally independent given $\mathbf{z}_{t-1}$; *ii) Sparse*: $z_{t,i}$ and $z_{t,j}$ are dependent given $\mathbf{z}_{t-1}$, but the latent Markov network $\mathcal{G}_{z_t}$ and time-lagged latent structure are sparse; *iii) Dense*: No sparsity restrictions on latent causal graph. Bold numbers indicate the best performance.

| Setting | Metric | CaDRe | iCITRIS | G-CaRL | CaRiNG | TDRL | LEAP | SlowVAE | PCL | i-VAE | TCL |
|---|---|---|---|---|---|---|---|---|---|---|---|
| **Independent** | MCC | **0.9811** | 0.6649 | 0.8023 | 0.8543 | 0.9106 | 0.8942 | 0.4312 | 0.6507 | 0.6738 | 0.5916 |
| | $R^2$ | **0.9626** | 0.7341 | 0.9012 | 0.8355 | 0.8649 | 0.7795 | 0.4270 | 0.4528 | 0.5917 | 0.3516 |
| **Sparse** | MCC | **0.9306** | 0.4531 | 0.7701 | 0.4924 | 0.6628 | 0.6453 | 0.3675 | 0.5275 | 0.4561 | 0.2629 |
| | $R^2$ | **0.9102** | 0.6326 | 0.5443 | 0.2897 | 0.6953 | 0.4637 | 0.2781 | 0.1852 | 0.2119 | 0.3028 |
| **Dense** | MCC | **0.6750** | 0.3274 | 0.6714 | 0.4893 | 0.3547 | 0.5842 | 0.1196 | 0.3865 | 0.2647 | 0.1324 |
| | $R^2$ | **0.9204** | 0.6875 | 0.8032 | 0.4925 | 0.7809 | 0.7723 | 0.5485 | 0.6302 | 0.1525 | 0.2060 |

## 5 Experiment

Based on the proposed framework, we conduct extensive experiments on both synthetic and real-world climate data to examine the identifiability of the latent process and observational causal graph, as well as climate forecasting and scientific interpretability in realistic climate systems.

### 5.1 On Synthetic Climate Data

**Baselines.** The data simulation processes and evaluation metrices are presented in Appendix C.1. In CD, we compare CaDRe with several constraint-based methods suited for nonparametric settings. Specifically, we include FCI [63] and CD-NOD [21], which handle latent confounders, and time-series methods PCMCI [60] and LPCMCI [15], which account for instantaneous and lagged effects with latent confounding. In CRL, we benchmark against CaRiNG [7], TDRL [75], LEAP [76], SlowVAE [28], PCL [24], i-VAE [26], TCL [23], and models that handle instantaneous effects, including iCITRIS [37] and G-CaRL [47]. Details are presented in Appendix C.1.

**Empirical Study.** We show performance on the CD and CRL in Table 2, and investigate different dimensionalities of observed variables. Our results on both latent representation learning metrics verify the effectiveness of our methodology under identifiabilty, and the result on $d_x = 100$ makes it scalable to high-dimensional data, if prior knowledge of the elimination of some dependences are provided by the physical law of climate [13] or LLM [42], supports our subsequent experiment on real-world data. Additionally, the study on different $d_z$ can be found in Appendix C.1.

**Comparison with Constraint-Based CD.** Figure 5 shows that CaDRe consistently outperforms all baselines across varying sample sizes, with performance improving as more data becomes available. In contrast, FCI performs poorly when latent confounders are dependent, often leading to low recall. CD-NOD relies on pseudo-causal sufficiency, assuming that latent variables are functions of surrogate variables, which does not hold in general latent settings. PCMCI ignores latent dynamics altogether, while LPCMCI assumes no causal relations among latent confounders, limiting its applicability in complex systems. These comparisons highlight the effectiveness of CaDRe in addressing the limitations of existing constraint-based methods.

**Comparison with Temporal CRL.** The MCC and $R^2$ results for the *independent* and *sparse* settings demonstrate that our model achieves component-wise identifiability (Theorem A.3). In contrast, other considered methods fail to recover latent variables, as they cannot properly address cases where the observed variables are causally-related. For the *dense* setting, our approach achieves monoblock identifiability (Theorem 1) with the highest $R^2$, while other methods exhibit significant degradation

Table 4: **Results on Temperature Forecasting.** Lower MSE/MAE is better. **Bold** numbers represent the best performance among the models, while underlined numbers denote the second-best.

| Dataset | Predicted Length | CaDRe | | TDRL | | CARD | | FITS | | MICN | | iTransformer | | TimesNet | | Autoformer | |
|---|---|---|---|---|---|---|---|---|---|---|---|---|---|---|---|---|---|
| | | MSE | MAE | MSE | MAE | MSE | MAE | MSE | MAE | MSE | MAE | MSE | MAE | MSE | MAE | MSE | MAE |
| CESM2 | 96 | 0.410 | 0.483 | 0.439 | 0.507 | **0.409** | **0.484** | 0.439 | 0.508 | 0.417 | 0.486 | 0.422 | 0.491 | 0.415 | 0.486 | 0.959 | 0.735 |
| | 192 | **0.412** | **0.487** | 0.440 | 0.508 | 0.422 | 0.493 | 0.447 | 0.515 | 1.559 | 0.984 | 0.425 | 0.495 | 0.417 | 0.497 | 1.574 | 0.972 |
| | 336 | **0.413** | **0.485** | 0.441 | 0.505 | 0.421 | 0.497 | 0.482 | 0.536 | 2.091 | 1.173 | 0.426 | 0.494 | 0.423 | 0.499 | 1.845 | 1.078 |

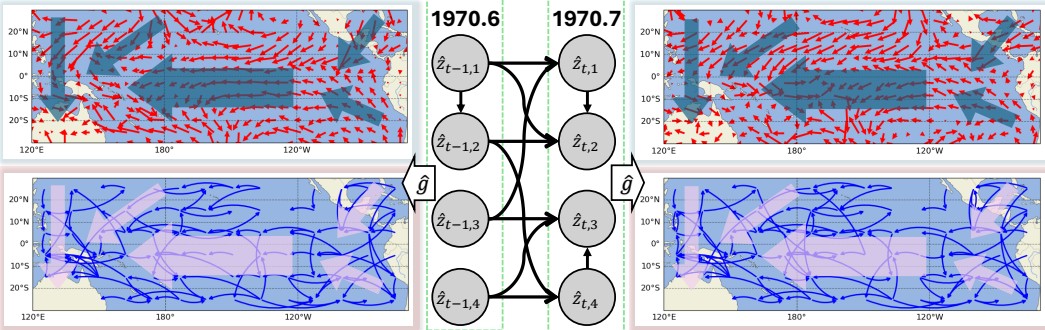

Figure 6: **Top:** Estimated instantaneous causal graph over climate grids. **Bottom:** Reference wind field from [53]. Blue arrows denote learned causal directions; red arrows indicate wind vectors.

because they are not specifically tailored to handle scenarios involving general noise in the generating function. These outcomes are consistent with our theoretical analysis.

## 5.2 On Real-World Climate Data

**Baselines.** Details about the climate datasets are presented in Appendix C.2. We consider the following state-of-the-art deep forecasting models for time series forecasting. First, we consider the conventional methods for time series forecasting, including Autoformer [72], TimesNet [71] and MICN [69]. Moreover, we consider several latest methods for time series analysis like CARD [70], FITS [73], and iTransformer [41]. Finally, we consider the TDRL [75]. We repeat each experiment over 3 random seeds and publish the average performance.

**Causal Discovery Consistency.** As the ground-truth causal graph is inaccessible in real climate data, we adopt the contemporaneous wind field [53] as a surrogate for evaluation. As shown in Figure 6, CaDRe recovers observational causal graphs closely consistent with physical wind patterns, serving as a scientific support. Specifically, CaDRe captures large-scale physical patterns (*e.g.*, westward flows in equatorial oceans, southwestward propagation near Central America), while revealing structurally complex zones along coastal boundaries. These dense, irregular edges may reflect coupled land–atmosphere dynamics or anthropogenic influences [68, 4]. The latent transition $\hat{\mathbf{z}}_{t-1} \to \hat{\mathbf{z}}_t$ is also visualized to unveil the hidden dynamic process in the scientific discovery.

**Weather Prediction.** We evaluate our method on the CESM2 sea surface temperature dataset for real-world temperature forecasting. As summarized in Table 4, our approach outperforms existing time-series forecasting models in precision, due to existing models struggling with causally-related observations and non-contaminated generation, restricting their usability in real-world climate data.

## 6 Conclusion

We focused on the causal understanding of climate science and proposed a causal model with latent processes and directly causally-related observed variables. We establish identifiability results and develop an estimation approach to uncovering the latent causal variables, latent causal process, and observational causal structures from the climate system, aiming to shed light on answering "why" questions in climate. Simulated experiments validate our theoretical findings, and real-world experiments offer causal insights for climate science.

**Limitations.** Our method shows performance degradation as the data dimensionality increases. A potential solution is to adopt a divide-and-conquer strategy by partitioning the variables into lower-dimensional subsets using prior geographical information.

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
