# OpenReview forum: "Learning General Causal Structures with Hidden Dynamic Process for Climate Analysis"
_NeurIPS.cc/2025/Conference — Submitted to NeurIPS 2025_

### Official Review · Reviewer_1X1S · 2025-07-01

**Clarity:** 2
**Significance:** 2
**Originality:** 2
**Rating:** 4
**Confidence:** 4

**Summary:**

This work establishes a refined connection between SEM (Structural Equation Model) with non-linear ICA (Independent Component Analysis), and applies it in climate analysis applications. Specifically, it extends ICA-based causality discovery to nonparametric settings with the presence of latent confounders. It proves an equivalence between SEM and non-linear ICA in this setting so that it is possible to learn an ICA instead of SEM. The training objective consists of ELBO and two penalty terms. Experiments on both synthetic and real-world climate datasets demonstrate the performance of proposed CaDRe and validate the theoretical design.

**Questions:**

Please refer to the weaknesses.

**Ethical Concerns:**

["NO or VERY MINOR ethics concerns only"]

**Final Justification:**

I appreciate the authors' rebuttal, which has addressed most of my concerns. I have adjusted my rating accordingly.

**Limitations:**

This is a pure ML research work without potential negative social impact.

**Quality:**

2

**Strengths And Weaknesses:**

This work is solid in general, with some (probably minor) issue before granting an acceptance. On the strengths, the proof of the existence of equivalent ICA to SEM is not trivial, and this work manages to make the formulations and proofs simple in presentation. Additionally, the performance of the proposed CaDRe could be validated on real-world datasets in the climate domain. The discovered of causailties also coincident with climate knowledge.

However, there are some weaknesses of this paper, from which the current manuscript can be improved.

W1. The proposed method could generalize to domains beyond climate data, wherever there could be latent variables, it seems CaDRe can be used. There are many time series data in other domains which could also serve as the testbed. Please correct me if there are some specific constraints that CaDRe can only be used on climate data.

W2. On synthetic data, the comparison are mainly with causal learning models, yet the comparison with time series models are not convincing, as the compared models PCMCI and LPCMCI are methods up to year 2020. Comparison of more recent time series models are needed.

W3. On real-world data, the experiment is conducted on only one dataset CESM2, whose statistics and scale is uncertain from the main page.

W3. It is not very clear how the latent variables (z) are interpreted for climate analysis. Specifically, there should be details of how visualization is conducted.

W4. The advantage of learning non-linear ICA instead of SEM should also be discussed and verified. In other words, this word should emphasize why bother to prove the existence of equivalent ICA and try to build it, instead of just training an SEM.

W5. Efficiency is not considered in this manuscript. The training/inference time is not reported for any method.

---

> ### Author Rebuttal · Authors · 2025-07-31
>
> Dear Reviewer 1X1S, we sincerely thank you for your valuable comments, constructive suggestions, and encouraging feedback. Below, we provide point-by-point responses and have updated the main paper and appendix accordingly.
>
> > W1: CaDRe could generalize to other time series domains
>
> Yes, you are correct. In light of your suggestions, we have evaluated CaDRe across 3 standard long-term time-series forecasting datasets from diverse domains, including finance (Exchange), public health (ILI), and traffic monitoring.
>
> As shown in the table below, CaDRe demonstrates consistently strong forecasting performance across a wide range of horizons, indicating robust generalization to varied time-series domains.
>
> |Dataset|I/O Len|CaDRe(MSE/MAE)|N-Transformer(MSE/MAE)|Autoformer(MSE/MAE)|MICN(MSE/MAE)|TimesNet(MSE/MAE)|
> |-------|------|-----------|-----------|-------------|-----------|-------------|
> |ILI|18-6|1.200/0.691|1.491/0.757|2.637/1.094|4.847/1.570|2.406/0.840|
> ||72-24|1.856/0.833|2.551/1.039|2.653/1.116|4.776/1.556|2.270/0.988|
> ||144-48|1.796/0.878|2.227/1.018|2.696/1.139|4.917/1.584|2.978/1.123|
> ||216-72|2.010/0.984|2.595/1.081|2.960/1.167|4.804/1.584|2.696/1.098|
> |ECL|18-6|0.114/0.216|0.134/0.242|0.136/0.254|0.250/0.338|0.128/0.236|
> ||72-24|0.121/0.220|0.140/0.246|0.144/0.257|0.258/0.342|0.134/0.242|
> ||144-48|0.124/0.225|0.155/0.260|0.163/0.275|0.271/0.353|0.149/0.256|
> ||216-72|0.131/0.232|0.169/0.274|0.175/0.287|0.279/0.357|0.166/0.271|
> |Traffic|18-6|0.487/0.307|0.797/0.347|0.554/0.322|0.475/0.287|0.781/0.337|
> ||72-24|0.452/0.303|0.625/0.319|0.508/0.318|0.454/0.276|0.608/0.307|
> ||144-48|0.412/0.282|0.574/0.314|0.497/0.319|0.450/0.275|0.553/0.296|
> ||216-72|0.400/0.278|0.593/0.325|0.524/0.330|0.473/0.287|0.564/0.303|
>
> > W2: Comparison of more recent time series causal discovery models in synthetic data
>
> Thanks for pointing this our. We additionally compare several recent time-series causal discovery methods, including TCDF [1] (2019), IDOL (2024), and TDRL (2022), on the same simulated datasets used in our main experiments.
>
> |Method|SHD↓|Precision↑|Recall↑|F1↑|
> |------|----|----------|-------|----|
> |CaDRe|0.185±0.021|0.803±0.037|0.830±0.012|0.815±0.025|
> |TCDF|0.429±0.035|0.442±0.041|0.384±0.046|0.411±0.014|
> |IDOL|0.297±0.029|0.624±0.053|0.598±0.040|0.610±0.021|
> |TDRL|0.348±0.031|0.505±0.037|0.462±0.014|0.483±0.056|
>
> **Implementation details**: We follow official implementations and default settings for all methods. For TCDF, we set the significance threshold to 0.9 and treat identified causes as parents. IDOL's latent graph is mapped to observed space and outputs instantaneous graphs. TDRL 's latent graph is mapped to observed space and outputs time-lagged graphs only.
>
> [1] Nauta, et al. "Causal discovery with attention-based convolutional neural networks." Machine Learning and Knowledge Extraction 1.1 (2019): 19.
>
> > W3 (1): CESM2 statistics is uncertain from the main page
>
> Thank you for pointing this out. The CESM2 Pacific SST dataset consists of monthly SST data from a 500-year pre-2020 control run, with 6000 time steps over ocean-only regions. It retains a native resolution of 186 × 151, totaling 28086 spatial points—24749 valid SST observations and 3337 land points excluded from analysis. For efficiency, we use a downsampled 6 × 14 grid (84 points). This is described in **Main Paper, Line 296** and **Appendix, Lines 1347–1355**, and has been further highlighted in the revision.
>
> > W3 (2): On real-world data, the experiment is conducted on only one dataset, CESM2
>
> Thank you for the suggestion. In addition to CESM2, we include two real-world datasets in our revised experiments:
>
> - **Weather** (see **Appendix, Table A10**), a standard benchmark with hourly meteorological observations;
> - **ERSST**, the NOAA Global Temperature Anomaly Dataset (1880–2025), consisting of 2052 monthly steps and 16,020 spatial grid points per step. We downscale it to 100 dimensions via spatial averaging for forecasting.
>
> Results on "CESM2", *Weather*, and *ERSST* are summarized below. CaDRe consistently achieves competitive or best performance in both MSE and MAE across various prediction lengths.
>
> |Dataset|Length|CaDRe MSE|CaDRe MAE|iTransformer MSE|iTransformer MAE|Autoformer MSE|Autoformer MAE|TimesNet MSE|TimesNet MAE|MICN MSE|MICN MAE|CARD MSE|CARD MAE|FITS MSE|FITS MAE|
> |---|---|---|---|---|---|---|---|---|---|---|---|---|---|---|---|
> |CESM2|96|0.410|**0.483**|0.422|0.491|0.959|0.735|0.415|0.486|0.417|0.486|**0.409**|0.484|0.439|0.508|
> |CESM2|192|**0.412**|**0.487**|0.425|0.495|1.574|0.972|0.417|0.497|1.559|0.984|0.422|0.493|0.447|0.515|
> |CESM2|336|**0.413**|**0.485**|0.426|0.494|1.845|1.078|0.423|0.499|2.091|1.173|0.421|0.497|0.482|0.536|
> |Weather|96|**0.157**|**0.203**|0.168|0.214|0.225|0.259|0.180|0.231|0.199|0.256|0.423|0.497|0.172|0.221|
> |Weather|192|**0.207**|**0.248**|0.193|0.241|0.354|0.348|0.212|0.265|0.238|0.298|0.482|0.544|0.216|0.260|
> |Weather|336|**0.270**|**0.314**|0.426|0.494|0.354|0.348|0.423|0.499|0.316|0.496|0.525|0.596|0.386|0.439|
> |ERSST|96|**0.145**|**0.268**|0.247|0.264|0.953|0.272|0.432|0.508|0.726|0.765|0.197|0.273|0.539|0.297|
> |ERSST|192|**0.208**|**0.307**|0.251|0.535|1.024|0.908|0.452|0.585|1.263|0.892|0.233|0.375|0.226|0.752|
> |ERSST|336|**0.305**|**0.361**|0.305|0.659|1.387|1.353|0.581|0.607|1.173|1.172|0.487|0.484|0.439|0.535|
>
> These additions strengthen the generalizability of our method on diverse real-world climate datasets. All experiments follow consistent preprocessing and evaluation protocols as detailed in the main paper.
>
> > W3 (3): It is not very clear how the latent variables (z) are interpreted for climate analysis
>
> Thank you for this important question. Latent factors are, by definition, **unobserved**, in principle, we cannot give a direct interpretation. This poses a central challenge in the field of CRL. However, once we are sure about the existence of latent factors and understand how it's related to measured variables, we can begin to interpret them and even come up with ways to measure them. For example, as long as we obtain domain knowledge of latent factors from climate experts/scientists, we can easily match them to the meaningful quantities, e.g., precipitation, solar radiation, or components of a climate foundational model representation. This mirrors historical processes in science, such as the discovery of viruses, which were first hypothesized based on indirect evidence and later confirmed and measured directly.
>
> We have included this perspective in our main paper and regard it as our future work.
>
> > W3 (4): There should be details of how visualization is conducted
>
> We clarify the visualization procedure as follows:
>
> - **Wind System Visualization**: The wind data consists of two components: the vertical component $u$ and the horizontal component $v$. For each spatial location $a$, we draw an arrow originating at $(\lambda_a, \phi_a)$, with its direction and length determined by the vector $(u_a, v_a)$: the direction indicates wind flow, and the length is proportional to the magnitude of the vector.
>
> - **Causal Graph Visualization**: If an edge from region $a$ to region $b$ in estimated causal adjacency matrix $B$ is present, i.e., the weight $B_{a,b}$ is nonzero, we draw an arrow $a \rightarrow b$, according to their positions in the dataset.
>
> > W4: Why train an ICA instead of SEM
>
> Thank you for this valuable question. The key distinction lies in the **nonparametric setting** of our SEM. In the **parametric additive noise setting**, where the SEM takes the form $X = f(X) + E$, the noise term can be isolated as $E = X - f(X)$, allowing a direct reconstruction-based loss $|X - f(X)|$ as used in prior work [2,3]. However, in the **nonparametric setting** $X = f(X, E)$, the noise $E$ cannot be separated from $X$ via subtraction. This makes direct optimization of the SEM via reconstruction loss ill-posed for causal discovery. In contrast, nonlinear ICA offers a principled alternative by enabling recovery of latent sources (here, $E$) under identifiability guarantees, thereby allowing recovery of the causal graph. We have clarified this point in the revised manuscript.
>
> [2] Zheng, Xun, et al. "Dags with no tears: Continuous optimization for structure learning." Advances in neural information processing systems 31 (2018).
>
> [3] Lachapelle, Sébastien, et al. "Gradient-based neural dag learning." arXiv preprint arXiv:1906.02226 (2019).
>
> > W5: Efficiency is not considered in this manuscript.
>
> Thank you for highlighting this important point. We agree that computational efficiency is an important consideration. In our revised manuscript, we have included measurements of training time, memory usage, and inference latency to provide a more complete assessment of efficiency.
>
> - **Climate Forecasting Model – Training and Inference Efficiency**
>   (CESM2, input dim = 82, sequence length = 96, batch size = 1; measured on NVIDIA A100-SXM4-80GB with CUDA 12.6, averaged over 100 runs):
>
> |Metric|CaDRe|Autoformer|TimesNet|CARD|MICN|FITS|TDRL|iTransformer|
> |------|-----|----------|--------|----|----|----|----|-------------|
> |**TrainingTime(s)**|613|487|12972|1513|1826|392|318|709|
> |**Memory(GB)**|1.234|1.843|5.517|2.018|1.045|0.886|1.027|1.093|
> |**InferenceLatency(ms)**|**1.095±0.203**|8.414±3.386|11.867±2.895|2.620±0.932|4.315±1.367|1.953±0.874|0.974±0.126|0.919±0.185|
>
> These results are shown in **Appendix, Fig. A8**, and are now explicitly referenced in the main text.
>
> - **Causal Discovery – Inference Time Comparison (ms)**
>   (Using official open-source implementations from *Tigramite* and *Causal-Learn*, which are training-free):
>
> |Method|CaDRe|FCI|CD-NOD|PCMCI|LPCMCI|PC|
> |------|-----|---|------|-----|------|---|
> |Latency|**1.10±0.20**|999.09±16.27|2242.88±27.14|3391.35±76.64|3508.50±123.36|2230.72±27.94|
>
> These results show that CaDRe is highly efficient in both training and inference, while maintaining strong performance in causal representation learning and discovery.

---

> ### Author Response · Authors · 2025-08-05
> **Kind Request for Reviewer 1X1S's Feedback**
>
> Dear Reviewer 1X1S,
>
> We are grateful for your time on our paper, your constructive comments, and your recognition of the significance and novelty of our work. Could you please have a look at our response and let us know whether your concerns have been addressed, regarding
>
> - **W1:** Generalization of CaDRe to other domains and additional experiments
> - **W2:** Comparison with more recent time series causal discovery models on synthetic data
> - **W3 (1,2):** CESM2 statistics and additional experiments on climate across **two new datasets**
> - **W3 (3,4):** Physical interpretability of latent variables and details on how the visualization is conducted
> - **W4:** The rationale for training an ICA model instead of directly training a SEM.
> - **W5:** Clarification of CaDRe’s efficiency in terms of training time, memory cost, and inference time.
>
> Additionally, for **W4**, we provide further empirical evidence and identifiability analysis to complement our initial response:
> ____
> > W4: Why train an ICA instead of SEM
>
> - **Experimental Verifications:** We tested direct SEM training by replacing **$s_t$ prior estimation** in CaDRe with the likelihood-based objective (if we assume a Gaussian distribution, it reduces to a reconstruction in [1]) as follows:
>
>
>     $$
>      \max \mathbb{E}\_{x_{t} \sim P_{x_{t}}} \sum_{j=1}^d \log p_j \left( x_{t,j} \mid x_{t, \pi_j}, z_{t} \right),$$
>
>     where $\pi_j$ denotes the set of parents of node $x_j$ in observational causal graph. This equation is a variant of Eq. (4) in [2] that accounts for latent confounders, denoted **CaDRe_SEM**. All other terms (e.g., DAG constraint, Jacobian-based edge support) follow our settings, with identical datasets, implementations, and hyperparameters.
>
>     |Method|$d_x$|SHD↓|Precision↑|Recall↑|F1↑|
>     |------|-----|----|----------|-------|----|
>     |**CaDRe_ICA**|3|0.00|1.00|1.00|1.00|
>     ||6|0.18|0.80|0.83|0.81|
>     ||8|0.29|0.76|0.78|0.77|
>     ||10|0.43|0.63|0.65|0.64|
>     |**CaDRe_SEM**|3|0.12|0.86|0.82|0.84|
>     ||6|0.40|0.64|0.60|0.62|
>     ||8|0.51|0.50|0.42|0.46|
>     ||10|0.56|0.49|0.41|0.44|
>
>
>     As shown, **CaDRe_SEM** shows a sharp performance decline, with substantially higher SHD and lower precision/recall, confirming that direct SEM training degrades in the nonparametric latent setting.
>
> - **ICA Establishes Identifiability:** Indeed, causal models to be identified here can be written as either an SEM or ICA. We reformulate the SEMs into a constrained form of nonlinear ICA primarily to *establish identifiability results in a natural way*, since no existing theory guarantees identifiability for nonparametric SEMs with latent variables.
>
> [1] Zheng, Xun, et al. "Dags with no tears: Continuous optimization for structure learning." Advances in neural information processing systems 31 (2018).
>
> [2] Lachapelle, Sébastien, et al. "Gradient-based neural dag learning." arXiv preprint arXiv:1906.02226 (2019).
> ___
> We hope these address your concerns. As the discussion phase ends in 2 days, we would greatly appreciate it if you could let us know if you have any remaining questions or suggestions.
>
> Your further feedback would be highly appreciated.
>
> Best,
>
> The Authors of Submission 9138

---

> > ### Comment · Reviewer_1X1S · 2025-08-07
> > **Thank you for the rebuttal!**
> >
> > I appreciate the authors' rebuttal, which has addressed most of my concerns. I will adjust my rating accordingly.

---

> > > ### Author Response · Authors · 2025-08-07
> > > **Appreciation for your time, comments, and support!**
> > >
> > > Dear Reviewer 1X1S,
> > >
> > > Thank you very much for taking the time to review our work. We are sincerely grateful for your constructive comments and are pleased to hear that our responses addressed your concerns.
> > >
> > > With best regards,
> > >
> > > The Authors of Submission 9138

---

### Official Review · Reviewer_NFts · 2025-07-01

**Clarity:** 2
**Significance:** 3
**Originality:** 2
**Rating:** 5
**Confidence:** 4

**Summary:**

This paper handles causal discovery over observational variables while identifying the latent variables. The authors assume a generative model grounded in climate analysis intuitions where an observed variable is causally affected not only by its lagged or instantaneous counterparts but also by some of the latent variables. The authors also assume that some of the latents $z_t$ are not necessarily causal parents but may modulate the causal mechanism behind observed variables $x_t$ through a random term $s_t$. The authors demonstrate a pointwise identification of latent variables and an “edge-by-edge” identification over the causal graph over observed variables. Experiments show improvement in structure recovery w.r.t. baselines.

**Questions:**

- Theorem 3 (lines 202–203): Definition 5 is missing from the paper (present only in the appendix), making the theorem statement on the identification of observational causal graph unclear. Moreover, the definition in the appendix refers to $s_t$, creating confusion because $s_t$ are the stochastic mediators. The authors did not claim such identifying $s_t$ in the main text. Looking at the proof, Definition 5 appears to be an intermediate step near the end of observational causal graph identification. Can you please clarify the result related to Def 5 and how it relates to identifying the observed causal graph?
- In Table 2: Can you further elaborate on the meaning of “the best-converged result per seed is selected to avoid local minima”?
- In Figure 5, “$d_x = 6$ and $d_z = 3$” are very low dimensions; How do the exact metrics reported in Figure 5 evolve as a function of $d_x$, including other baselines?
- Figure 6: “CaDRe matches observational causal graphs to physical wind patterns.” What about other baselines that could provide the same instantaneous causal graph? How do they qualitatively match the physical wind patterns?

**Ethical Concerns:**

["NO or VERY MINOR ethics concerns only"]

**Final Justification:**

The authors addressed the main points I mentioned in the Weaknesses and Questions sections. The only point I am not convinced of is the response related to the choice of the best-converged result per seed. I think this is a concern general to many papers where more effort should be put into dealing with the impact of initialization on model performance, especially in this field. This does not, however, impede my ability to raise my score.

**Limitations:**

* Authors did not discuss how the developed approach may or may not generalize to other domains beyond climate analysis.
* The distributional assumptions like A2 and A3 are very strong.

**Quality:**

3

**Strengths And Weaknesses:**

## Strengths
- The authors made commendable efforts in clearly communicating their works, although some parts might have been better written.
- The theoretical part has discussed the main important aspects of the problem.
- The model proposed was compared against a very fair number of baselines in both causal discovery and time series forecasting.
- I took note of the quality of the appendix complementing the main paper (although some elements should have been stated in the core paper for better clarity—see below).

## Weaknesses
- For “$s_{t,i} = g_{si}(z_t, \epsilon_{x_{t,i}})$” in Eq. 1, authors mentioned “noise conditioned on $z_t$”; however, it should not be considered “noise,” since it depends on $z_t$. $s_t$ plays the role of an endogenous mediator.
- “$s_{t}$ is said to be ‘designed to capture inherent climatic variability, such as perturbations introduced by human activities on CO₂’”: I think further attention should be paid to explaining the idea behind $s_t$. I am not an expert in climate analysis, but it is unclear how $s_t$, being—for example—interpreted as perturbations induced by human activity on CO₂, could be “driven” by latent causes like solar radiation as said in the text.
- The authors mention causal representation learning, and in Table 1 the model CADRe claims to recover a “latent causal graph.” However, there is no discussion of the identification of the causal graphs between the latent variables (lagged and instantaneous edges). Unless mistaken, there is no clear discussion of how such a graph is identified or learned—only pointwise identifiability of latents in Thm 1. This should be clarified, plus I think it should be clarified that identifying $\mathrm{pa}_\ell(x_{i,t})$, although concerning latent variables, is not a classical task of CRL.
- “We denote $J_d(\hat z_{t+1})$ as the Jacobian matrix of the function $r$,” the notation is confusing: the Jacobian of $r$ should better be denoted $J_r$, but $r$ in line 235 has two arguments $(\hat z_{t+1}, \hat z_t)$. Can you please clarify the notations and enhance the presentation quality of the paragraph “Prior Estimation of $z_t$ and $s_t$”?


## Notation and Typo issues

- “the stochasticility” should be “stochasticity.”
- Bad notation to use $M$ for a functional in “A4 (Differentiability:)” (line 137).
- “As depicted in Eq. (A19),” in line 151, the equation is missing (I know it is in the separate appendix, but its omission disrupts the flow).
- Line 139: “where $h$ is differentiable.” should read “where $h_z$ is differentiable.”
-  Paragraph “Prior Estimation of $z_t$ and $s_t$.” is poorly written and difficult to read.
- Line 242: “we using” should be “we are using.”
-  Line 257: “$D(A)$” overlaps with $D$ used earlier for KL divergence.
-  Equations 10–11: Please specify the summands in $\Sigma$; it is very unclear.
- Line 267: Metric names should be at least mentioned, and definitions could be deferred to the appendix.

---

> ### Author Rebuttal · Authors · 2025-07-31
>
> Dear Reviewer NFts, thank you for your insightful feedback. Your comments helped improve the rigor of our work, particularly in notation, terminology, comparisons, and quantitative analysis. We sincerely appreciate your time and effort.
>
> We provide point-by-point responses to your comments below and have updated the manuscript accordingly.
>
> > W1: $s_{t,i}$ should be considered as an endogenous mediator
>
> Thank you for this valuable suggestion. Describing $s_{t,i}$ as an endogenous mediator of $z_t$ is more accurate than the term nonstationary noise, which we used following prior work [1]. We have revised the manuscript accordingly.
>
> [1] Huang et al., "Causal discovery and forecasting in nonstationary environments with state-space models."
>
> > W2: The scientific idea behind modeling $s_t$
>
> $s_t$ is **not limited** to human-induced effects but broadly captures uncertainties such as environmental variability and measurement noise. These can be modulated by latent dynamics, e.g., variance of $s_t$ is time-varying [2]. We have clarified this general interpretation in the revised manuscript.
>
> [2] Lashgari, et al. "Evaluation of simulated responses to climate forcings: a flexible statistical framework using confirmatory factor analysis and structural equation modelling–Part 1: Theory."
>
> > W3(1): No discussion of the identification of the latent causal graphs
>
> Thanks a lot for raising this point! The latent graph is identifiable up to permutation under the assumptions of sparse latent dynamics and sufficient variability, as discussed in **Main Paper, Line 159** and **Appendix, Section A.3**. In light of your suggestion, we have highlighted it in the main text.
>
> > W3(2): Identifying $pa_{O}(x_{t,i})$ is not a classical task of CRL
>
> You are totally right! CRL is a specific task within the broader domain of CD, concerned with learning causally related latent representations. To clarify, identifying $pa_{O}(x_{t,i})$ is the CD with latent variables. We have distinguished these two tasks throughout the revised manuscript.
>
> > W4(1): Notation on the Jacobian matrix, $J_r$ should has two arguments
>
> We appreciate your efforts in helping us improve the readability. We clarify that the notation $J_r(\hat{z}_t)$ denotes the Jacobian *w.r.t.* $\hat{z}_t$, rather than a function taking $\hat{z}_t$ as an input.
>
> > W4(2): Enhance the presentation quality of “Prior Estimation of $z_t$ and $s_t$"
>
> We now provide a clearer version in brief:
>
> ""
>
> To estimate priors that preserve causal structure, we minimize the KL divergence between the approximate posterior and a learned prior. Since the true prior is unknown, we use a normalizing flow conditioned on selected inputs. Input selection is guided by learned inverse transition functions $r_ i$, where $\hat{\epsilon}^z_ {t,i} = r_i(\hat{z}_ {t-1}, \hat{z}_ t)$ identifies the latent variables that causally influence $z_ {t,i}$. Formally, the Jacobian of the transformation $\kappa: \{\hat{z}_ {t-1}, \hat{z}_ t\} \rightarrow \{\hat{z}_ {t-1}, \hat{\epsilon}^z_ t\}$ encodes these dependencies:
> $
> J_ {\kappa}(\hat{z}_ {t-1}, \hat{z}_ t) =
> \begin{pmatrix}
>     \mathbf{I} & 0 \\\\
>     J_ r(\hat{z}_ {t-1}) & J_ r(\hat{z}_ t)
> \end{pmatrix},
> $
> where $J_ r(\hat{z}_ t)$ captures instantaneous structure and $J_ d(\hat{z}_ {t-1})$ captures time-lagged effects.
>
> A similar process is applied to observed variables via $w_i(\hat{z}_t, \hat{s}_t)$. Finally, we enforce conditional independence across components by minimizing KL divergences from the estimated $\hat{\epsilon}^z_t$ and $\hat{\epsilon}^x_t$ to standard Gaussian. This enables principled prior learning and reveals causal relations among $z_t$.
>
> ""
>
> The updates for W4 have been incorporated into our revised manuscript.
>
> > Notation and Typo issues
>
> Thank you for your detailed and attentive review. We have corrected all noted issues, including typos (e.g., “stochasticility”), unclear notation (e.g., $\bar{M}$, $D(A)$), missing references (e.g., Eq. A19), and phrasing errors (e.g., Lines 139, 242). We have also revised the paragraph on prior estimation and clarified Eq. (10)–(11) and metric definitions for better readability.
>
> > Q1: The motivation of Def 5 and how it relates to identifying the observed causal graph
>
> Thank you for this question. To motivate Def. 5, consider the linear case $x_t = B x_t + s_t$, where $s_t = (I - B)^{-1} x_t = M x_t$. If $s_t$ is identified component-wise without permutation (as required in Def. 5), then the support of $M$ is also identified, enabling recovery of $B = I - M^{-1}$. Any permutation in $s_t$ would corrupt $M$ and render $B$ unidentifiable.
>
> Our nonlinear setting follows the same principle: $s_t$ is identified component-wise via nonlinear ICA without permutation (conditioned on $z_t$), and the Jacobian replaces $M$ and $B$ to recover the causal graph over $x_t$ (see Main Paper, Lines 204–210).
>
> > Q2: Elaborate on the meaning of "best-converged result"
>
> Thank you for the question. Due to non-convexity in structure learning [3], for the same dataset, we run multiple trials with different initializations and report the one with the lowest total loss, referred to as the best-converged result.
>
> [3] Ng, Ignavier, Biwei Huang, and Kun Zhang. "Structure learning with continuous optimization: A sober look and beyond."
>
> > Q3: Baseline comparison with dimension evolving as a function of $d_x$
>
> Thank you for your thoughtful question. To address this, we report detailed results illustrating how the metrics in **Figure 5** evolve with increasing dimensionality $d_x = \{3,6,8,10\}$, while fixing the number of latent variables $d_z = 3$ and the number of samples $n = 10000$.
>
> The table below presents the SHD, Precision, Recall, and F1 score for CaDRe and four baselines across varying values of $d_x$:
>
> |Method|$d_x$|SHD↓|Precision↑|Recall↑|F1↑|
> |------|-----|----|----------|--------|----|
> |**CaDRe**|3|0.000|1.000|1.000|1.000|
> ||6|0.185|0.803|0.830|0.815|
> ||8|0.295|0.761|0.789|0.778|
> ||10|0.432|0.638|0.656|0.643|
> |**FCI**|3|0.186|0.801|0.760|0.780|
> ||6|0.384|0.476|0.394|0.431|
> ||8|0.447|0.398|0.321|0.356|
> ||10|0.492|0.355|0.284|0.315|
> |**CDNOD**|3|0.163|0.821|0.782|0.801|
> ||6|0.452|0.432|0.419|0.425|
> ||8|0.509|0.365|0.312|0.336|
> ||10|0.546|0.328|0.276|0.300|
> |**PCMCI**|3|0.139|0.843|0.803|0.822|
> ||6|0.431|0.488|0.386|0.430|
> ||8|0.501|0.397|0.308|0.347|
> ||10|0.548|0.365|0.284|0.319|
> |**LPCMCI**|3|0.116|0.864|0.823|0.843|
> ||6|0.337|0.637|0.621|0.629|
> ||8|0.441|0.535|0.486|0.509|
> ||10|0.487|0.482|0.432|0.456|
>
> We have included a line plot in the revised version.
>
> > Q4 (1): Can other baselines that could provide the same instantaneous causal graph
>
> Thank you for the insightful question. While baselines such as FCI and PCMCI can estimate instantaneous graphs, their outputs tend to be noisier and less aligned with known physical structures. For completeness, we have added their visualized comparisons in the appendix.
>
> > Q4 (2): Qualitatively match the physical wind patterns
>
> To assess alignment with physical wind patterns, we propose two metrics, **Wind-SHD (WSHD)** and **Wind-TPR (WTPR)**: WSHD measures the normalized SHD between the estimated graph $B$ and the wind-induced reference graph $B_{\text{ref}}$, while WTPR computes the recall of edges in $B$ w.r.t. $B_{\text{ref}}$.
>
> We report results below. These results demonstrate that CaDRe achieves the best alignment with the wind-induced causal graph.
>
> |Metric|CaDRe|FCI|CDNOD|PCMCI|LPCMCI|
> |------|-----|---|-----|-----|-------|
> |WSHD↓|0.012|0.028|0.031|0.024|0.019|
> |WTPR↑|0.532|0.236|0.251|0.198|0.274|
>
> > L1: How the developed approach may or may not generalize to other domains
>
> Thank you for giving us the opportunity to clarify the generalization capability of CaDRe. While our focus is climate data, CaDRe generalizes well to other time-series domains with latent variables. We evaluate it on 3 standard benchmarks: **finance** (Exchange), **public health** (ILI), and **traffic monitoring** (Traffic), using varied input/output lengths.
>
> As shown below, CaDRe consistently outperforms or matches strong baselines across domains, demonstrating robust generalization and applicability beyond climate. We have added this part in our revised manuscript.
>
> |Dataset|I/O Len|CaDRe(MSE/MAE)|N-Transformer(MSE/MAE)|Autoformer(MSE/MAE)|MICN(MSE/MAE)|TimesNet(MSE/MAE)|
> |-------|------|-----------|-----------|-------------|-----------|-------------|
> |ILI|18-6|1.200/0.691|1.491/0.757|2.637/1.094|4.847/1.570|2.406/0.840|
> ||72-24|1.856/0.833|2.551/1.039|2.653/1.116|4.776/1.556|2.270/0.988|
> ||144-48|1.796/0.878|2.227/1.018|2.696/1.139|4.917/1.584|2.978/1.123|
> ||216-72|2.010/0.984|2.595/1.081|2.960/1.167|4.804/1.584|2.696/1.098|
> |ECL|18-6|0.114/0.216|0.134/0.242|0.136/0.254|0.250/0.338|0.128/0.236|
> ||72-24|0.121/0.220|0.140/0.246|0.144/0.257|0.258/0.342|0.134/0.242|
> ||144-48|0.124/0.225|0.155/0.260|0.163/0.275|0.271/0.353|0.149/0.256|
> ||216-72|0.131/0.232|0.169/0.274|0.175/0.287|0.279/0.357|0.166/0.271|
> |Traffic|18-6|0.487/0.307|0.797/0.347|0.554/0.322|0.475/0.287|0.781/0.337|
> ||72-24|0.452/0.303|0.625/0.319|0.508/0.318|0.454/0.276|0.608/0.307|
> ||144-48|0.412/0.282|0.574/0.314|0.497/0.319|0.450/0.275|0.553/0.296|
> ||216-72|0.400/0.278|0.593/0.325|0.524/0.330|0.473/0.287|0.564/0.303|
>
> > L2: The distributional assumptions like A2 and A3 are very strong.
>
> We agree that A2 and A3 are strong in an absolute sense, though they are standard in nonparametric identifiability. However, they are **weaker than assumptions in prior CRL works** compared to prior CRL works. In previous nonlinear-ICA-based CRL, which often require **invertibility**. As shown in Appendix A.8, invertible functions form a **zero-measure** subset of injective operators (A2), making A2 less restrictive. Latent drift (A3) is also weaker than invertibility and can hold under heteroskedastic noise. Both reflect distributional variability, which is plausible in dynamic climate data. We have clarified this in the revised manuscript.

---

> > ### Comment · Reviewer_NFts · 2025-08-04
> >
> > Dear authors, thank you for your detailed response. You have addressed most of my concerns. The only point I am not convinced of is the response related to the choice of the best-converged result per seed. I think this is a concern general to many papers where more effort should be put into dealing with the impact of initialization on model performance, especially in this field. This does not, however, impede my ability to raise my score.

---

> > > ### Author Response · Authors · 2025-08-05
> > > **Thank you once again for your valuable comments and suggestions!**
> > >
> > > Dear Reviewer NFts,
> > >
> > > Thank you once again for your time dedicated to reviewing this paper and further engagement and insight. We appreciate you pointing out an open problem involved in training many deep learning models, which deserves more attention in future research, and we are grateful for your understanding.
> > >
> > > With best wishes,
> > >
> > > The Authors of Submission 9138

---

### Official Review · Reviewer_G8Kr · 2025-07-02

**Clarity:** 3
**Significance:** 3
**Originality:** 3
**Rating:** 4
**Confidence:** 4

**Summary:**

This paper introduces CaDRe, which aims to uncover the latent driving forces and causal relations among the observed variables in climate analysis. They provided a detailed theoretical analysis and demonstrated the effectiveness of the proposed method using synthetic and real-world experiments.

**Questions:**

1. The author maps observed variables to latent variables, and the dimension between variables changes from $d_x$ to $d_z$. It is expected that latent variables can represent variables with clear meanings, such as pressure and precipitation, but is this representation consistent? Is the physical meaning of latent variables fixed and clear across different datasets?

2. We can understand that due to the rapid changes in the climate system, the time lagged effects between variables may not need to be set to a particularly high order, but using only 1 lag is not sufficient. We can provide some experimental results under high-order conditions.

3. The experimental results in Table 2 show that when $d_x$ becomes larger, the effect of the fixed 3-dimensional latent variables $d_z$ becomes worse. However, we noticed that when $d_x$ is 100, $d_z$ is also 3, but its index results are significantly better. How should we understand this?

**Ethical Concerns:**

["NO or VERY MINOR ethics concerns only"]

**Final Justification:**

I have checked the author's rebuttal and other reviewers' comments, and I still tend to accept this paper.

**Limitations:**

yes.

**Quality:**

3

**Strengths And Weaknesses:**

**Strengths**

1. This paper provides novel methods and insights for the application of causal discovery in the field of climate analysis, and achieves solid results.

2. This paper provides a detailed theoretical analysis of the proposed causal discovery framework.

**Weaknesses**

1. This paper uses an MLP to map observed variables into a latent space for causal discovery. However, the paper does not explain whether the latent space has a clear physical meaning and whether it has a consistent meaning across different datasets.

2. Although the authors mention discussions about higher-order Markov structures in the article, there does not seem to be much experimental verification.

---

> ### Author Rebuttal · Authors · 2025-07-31
>
> Dear reviewer G8Kr, we are very grateful for your valuable comments, helpful suggestions, and encouragement. We provide the point-to-point response to your comments below and have updated the paper and appendix accordingly.
>
> > W1 & Q1: Physical meaning of the latent space
>
> Thank you for this important question. Latent factors are, by definition, **unobserved**, in principle, we cannot give a direct interpretation. This poses a central challenge in the field of CRL. However, once we are sure about the existence of latent factors and understand how it's related to measured variables, we can begin to interpret them and even come up with ways to measure them. For example, as long as we obtain domain knowledge of latent factors from climate experts/scientists, we can easily match them to the meaningful quantities, e.g., precipitation, solar radiation, or components of a climate foundational model representation. This mirrors historical processes in science, such as the discovery of viruses, which were first hypothesized based on indirect evidence and later confirmed and measured directly.
>
> We have included this perspective in our main paper and regard it as our future work.
>
> > W1(2): Whether it has a consistent meaning across different datasets
>
> Thank you for raising this issue. For different climate subsystems, latent variables may not share consistent physical meanings across datasets, as observed variables and measurement settings differ. Each dataset yields a distinct observational causal graph governed by its latent confounders; thus, the underlying latent factors are necessarily different.
>
> However, for datasets describing the same climate subsystem, despite differences in measurements due to sampling, the recovered latent factors might consistently represent the same underlying phenomena and complement each other.
>
> We have included this discussion in the revised manuscript.
>
> > W2 & Q2: Experimental verifications on the higher-order Markov structure
>
> Thank you for the valuable suggestions! To verify our framework’s capability on higher-order latent dynamics, we conduct experiments using a **second-order latent Markov process**. Results are reported below:
>
> | $d_z$ | $d_x$   | SHD $(J_{\hat{g}}(\hat{x}_t))$ | TPR             | Precision        | MCC $(s_t)$        | MCC $(z_t)$        | SHD $(J_r(\hat{z}_t))$ | SHD $(J_d(\hat{z}_{t-1}))$ | $R^2$            |
> |:-----:|:-------:|:-----------------------------:|:---------------:|:----------------:|:------------------:|:------------------:|:----------------------:|:---------------------------:|:----------------:|
> | 3     | 3       | $0.31 \pm 0.01$    | $0.91 \pm 0.02$ | $0.93 \pm 0.03$ | $0.9780 \pm 0.01$   | $0.9825 \pm 0.01$   | $0.26 \pm 0.06$         | $0.30 \pm 0.04$              | $0.93 \pm 0.04$  |
> | 3     | 6       | $0.19 \pm 0.07$               | $0.81 \pm 0.04$ | $0.79 \pm 0.08$  | $0.9560 \pm 0.03$   | $0.9520 \pm 0.01$   | $0.25 \pm 0.05$         | $0.34 \pm 0.08$              | $0.91 \pm 0.02$  |
> | 3     | 8       | $0.27 \pm 0.06$               | $0.80 \pm 0.04$ | $0.70 \pm 0.05$  | $0.9040 \pm 0.09$   | $0.9610 \pm 0.10$   | $0.34 \pm 0.09$         | $0.32 \pm 0.10$              | $0.93 \pm 0.03$  |
> | 3     | 10      | $0.45 \pm 0.06$               | $0.64 \pm 0.10$ | $0.62 \pm 0.13$  | $0.8470 \pm 0.06$   | $0.9630 \pm 0.03$   | $0.30 \pm 0.06$         | $0.39 \pm 0.06$              | $0.91 \pm 0.10$  |
> | 3     | $100^*$ | $0.18 \pm 0.03$               | $0.81 \pm 0.04$ | $0.80 \pm 0.03$  | $0.9100 \pm 0.03$   | $0.9570 \pm 0.02$   | $0.22 \pm 0.02$         | $0.28 \pm 0.09$              | $0.92 \pm 0.13$  |
>
> The table shows that CaDRe maintains high CRL quality and accurate causal discovery, confirming its effectiveness under higher-order latent dynamics.
>
> All other settings are aligned with those reported in the main paper. This setting allows us to evaluate how model performance changes under higher-order latent dynamics. For simulating datasets, the latent process is simulated using a leaky non-linear autoregressive model with $L=2$:
>
> $$
>     z_t = (I - B^{-1}) \left( \sigma\left( \sum_{\ell=1}^{L} \mathbf{W}^{(\ell)} z_{t - \ell} \right) + \boldsymbol{\epsilon}^z_t \right), \quad \boldsymbol{\epsilon}^z_t \sim \mathcal{N}(0, \sigma_z^2 \mathbf{I}),
> $$
>
> where $\sigma(\cdot)$ is leaky ReLU, and $\mathbf{W}^{(\ell)}$ are lag-$\ell$ transition matrices, $B$ is the instantanous latent causal adjacency matrix.
>
> > Q3: How to understand $d_x=100$ performs significantly better $d_x=10$
>
> Thank you for pointing this out. The key reason is that we incorporated a physical prior in the $d_x = 100$ experiment (marked with “$^*$”), but did not include this prior in the lower-dimensional cases ($d_x = 3, 6, 8, 10$), as described in **Main Paper, Lines 275–278** and **Appendix, Lines 1283–1287**.
>
> This prior reflects a physical assumption in climate systems: instantaneous interactions between distant spatial regions are unlikely. Accordingly, we removed 75% of the edges from the fully connected graph during initialization to enforce sparsity in $d_x=100$. To further clarify the role of this prior and our setup, we include an **ablation study** below, which demonstrates that incorporating the physical prior significantly improves performance, particularly in causal discovery.
>
> | $d_z$ | $d_x$             | SHD $(J_{\hat{g}}(\hat{x}_t))$ | TPR             | Precision        | MCC $(s_t)$        | MCC $(z_t)$        | SHD $(J_r(\hat{z}_t))$ | SHD $(J_d(\hat{z}_{t-1}))$ | $R^2$            |
> |:-----:|:-----------------:|:-----------------------------:|:---------------:|:----------------:|:------------------:|:------------------:|:----------------------:|:---------------------------:|:----------------:|
> | 3     | 10 (w/o prior)    | $0.45 \pm 0.06$               | $0.64 \pm 0.10$ | $0.62 \pm 0.13$  | $0.8470 \pm 0.06$   | $0.9630 \pm 0.03$   | $0.30 \pm 0.06$         | $0.39 \pm 0.06$              | $0.91 \pm 0.10$  |
> | 3     | 10 (w/ prior)     | $\mathbf{0.28 \pm 0.04}$      | $\mathbf{0.76 \pm 0.05}$ | $\mathbf{0.74 \pm 0.07}$  | $\mathbf{0.9010 \pm 0.03}$   | $\mathbf{0.9670 \pm 0.02}$   | $\mathbf{0.26 \pm 0.04}$         | $\mathbf{0.31 \pm 0.05}$              | $\mathbf{0.94 \pm 0.04}$  |
> | 3     | 100 (w/o prior)   | $\mathbf{0.38 \pm 0.08}$      | $\mathbf{0.61 \pm 0.06}$ | $\mathbf{0.57 \pm 0.09}$  | $\mathbf{0.8330 \pm 0.07}$   | $\mathbf{0.9380 \pm 0.05}$   | $\mathbf{0.36 \pm 0.10}$         | $\mathbf{0.42 \pm 0.07}$              | $\mathbf{0.86 \pm 0.12}$  |
> | 3     | $100^*$ (w/ prior)| $0.18 \pm 0.03$               | $0.81 \pm 0.04$ | $0.80 \pm 0.03$  | $0.9100 \pm 0.03$   | $0.9570 \pm 0.02$   | $0.22 \pm 0.02$         | $0.28 \pm 0.09$              | $0.92 \pm 0.13$  |
>
> We have now explicitly clarified this difference in the revised manuscript to avoid confusion.

---

> > ### Author Response · Authors · 2025-08-05
> > **Have the concerns been adequately addressed in the response?**
> >
> > Dear Reviewer G8Kr,
> >
> > Thank you so much for your detailed and constructive review. We are truly grateful for your recognition of the significance and novelty of our work in both causal discovery and climate science.
> >
> > In our rebuttal, we addressed your comments point by point, covering:
> >
> > - **W1 & Q1** Clarification of the physical interpretability of latent variables.
> > - **W1 (2)** Discussion on the consistency of the physical meaning of latent variables across different climate datasets.
> > - **W2 & Q2** Experimental verifications of the higher-order Markov structure
> > - **Q3** Explaination for why $d_x = 100$ with physical prior performs better than $d_x=6$
> >
> > We hope these address your concerns. As the discussion phase ends in 2 days, we would greatly appreciate it if you could let us know if you have any remaining questions or suggestions.
> >
> > Thank you again for your time and thoughtful feedback!
> >
> > Best,
> >
> > The Authors of Submission 9138

---

> > > ### Comment · Reviewer_G8Kr · 2025-08-08
> > >
> > > I have checked the authors' rebuttal, which has addressed my concerns. Then I still tend to accept this paper.

---

> ### Author Response · Authors · 2025-08-08
> **Thank you very much!**
>
> Dear Reviewer G8Kr,
>
> Thank you very much for taking the time to review our work. We are sincerely grateful for your constructive comments.
>
> We feel delighted that your concerns have been fully addressed. We would be grateful if you might consider updating your rating accordingly. Thank you again for your thoughtful and invaluable feedback!
>
> With best wishes,
>
> The Authors of Submission 9138

---

### Official Review · Reviewer_p8vt · 2025-07-03

**Clarity:** 2
**Significance:** 2
**Originality:** 2
**Rating:** 3
**Confidence:** 2

**Summary:**

This paper, introduces a novel framework called CaDRe (Causal Discovery and Representation Learning). Its primary goal is to uncover complex causal structures within climate systems from purely observational time-series data. A core innovation of CaDRe lies in its unification of Causal Representation Learning (CRL) and Causal Discovery (CD), establishing rigorous identifiability conditions for both latent processes and observed causal graphs, even without strong parametric assumptions. The framework leverages a novel theoretical connection between Structural Equation Models (SEM) and Nonlinear Independent Component Analysis (ICA) to achieve this. Methodologically, CaDRe is instantiated as a state-space Variational Autoencoder (VAE) that incorporates flow-based priors and gradient-based structural penalties to ensure the identifiability and sparsity of the learned causal graphs.
Empirical validation, through synthetic data experiments, confirms its theoretical claims, demonstrating CaDRe's ability to recover latent representations and causal structures. On real-world climate datasets (CESM2 sea surface temperature), CaDRe achieves competitive forecasting performance and visualizes causal graphs consistent with domain knowledge, such as wind circulation patterns and land-sea interactions, even revealing structural patterns that may inspire new hypotheses in climate science.

**Questions:**

- How robust is CaDRe when some of the conditions in Assumptions A1-A5 are partially violated in real-world data? Are there ways to quantify the degree of these violations and assess their impact on the results?
- How can the "physical interpretability" of the latent variables $$z_t$$ and noise terms $$s_t$$ be quantified or evaluated beyond consistency with domain knowledge? Are there more rigorous metrics?

**Ethical Concerns:**

["NO or VERY MINOR ethics concerns only"]

**Limitations:**

- Performance Degradation with Dimensionality: The method shows reduced performance as d increases, though partitioning variables via geographical priors is suggested as a solution.
- Reliance on Observational Data: The framework assumes access to time-series data, but climate observations are often sparse or noisy, which may impact identifiability in practice.

**Quality:**

2

**Strengths And Weaknesses:**

Strengths
- the paper explicitly identifies the need for a "unified framework". By simultaneously addressing both latent variable discovery (CRL) and observed variable causal discovery (CD), CaDRe tackles a more complete and realistic problem than methods that focus on only one aspect or assume simpler data-generating processes.
- The paper provides rigorous identifiability proofs in a nonparametric setting , which is crucial for the complex and often unknown functional forms in climate systems.The assumptions are clearly stated and discussed, indicating a deep theoretical understanding.
- Beyond competitive forecasting performance , CaDRe's ability to generate "visualized causal graphs consistent with domain knowledge" is a major strength for scientific applications.
Weaknesses:
- The method exhibits performance degradation as the data dimensionality increases. This implies that as the number of variables in climate models grows, or when analyzing at finer scales where more interconnected factors need to be considered, CaDRe's efficiency and accuracy might be impacted. This could limit its direct applicability in handling extremely complex and high-dimensional climate datasets.

---

> ### Author Rebuttal · Authors · 2025-07-31
>
> Dear Reviewer p8vt, thank you for your constructive comments. Your insights have helped us significantly improve the clarity, empirical validation, and theoretical framing of our work. Below, please see our point-to-point responses.
>
> > W1: Performance/efficiency degrades with dimensionality in synthetic data.
>
> That is a great point! We sincerely appreciate the insightful comment about the importance of evaluating the scalability of our approach. Although the difficulty of causal process identification increases with dimensionality, some identifications can still be achieved by using a climate prior. Please kindly find that we have included experiments with $d_x = 100^*$ in **Main Paper, Table 2** as follows:
>
> | $d_z$ | $d_x$  | SHD $(J_{\hat{g}}(\hat{x}_t))$ | TPR             | Precision       | MCC $(s_t)$        | MCC $(z_t)$        | SHD $(J_r(\hat{z}_t))$ | SHD $(J_d(\hat{z}_{t-1}))$ | $R^2$            |
> |:-----:|:------:|:-----------------------------:|:---------------:|:----------------:|:------------------:|:------------------:|:----------------------:|:---------------------------:|:----------------:|
> | 3     | 100*   | $0.17 \pm 0.02$               | $0.80 \pm 0.05$ | $0.81 \pm 0.02$  | $0.9131 \pm 0.02$   | $0.9565 \pm 0.02$   | $0.21 \pm 0.01$         | $0.29 \pm 0.10$              | $0.93 \pm 0.03$  |
>
> To overcome this challenge, in this setting (*), we mask 75% of edges in the initial fully connected graph based on **spatial distance**, under the assumption that distant regions do not directly interact, which is aligned with domain knowledge in climate systems. This prior reduces spurious dependencies and mitigates local minima during optimization, as described in **Main Paper, Line 276 - 278** and **Appendix, 1283-1287**. We have emphasized this in our revised manuscript.
>
> To further validate robustness to dimensionality, we conduct additional experiments with $d_x \in \{20, 50, 80, 100, 200\}$, applying the same physical prior in each case. As shown in the table below, our method maintains strong performance across all metrics, with only mild degradation as dimensionality increases. Inference time remains efficient due to nonlinear ICA-based structure learning is equivalent to a one-step generation.
>
> | $d_z$ | $d_x$  | SHD $(J_{\hat{g}}(\hat{x}_t))$ | TPR             | Precision       | MCC $(s_t)$        | MCC $(z_t)$        | SHD $(J_r(\hat{z}_t))$ | SHD $(J_d(\hat{z}_{t-1}))$ | $R^2$            | Inference Time (ms) |
> |:-----:|:------:|:-----------------------------:|:---------------:|:----------------:|:------------------:|:------------------:|:----------------------:|:---------------------------:|:----------------:|:--------------------:|
> | 3     | 20     | $0.09 \pm 0.01$               | $0.92 \pm 0.02$ | $0.89 \pm 0.01$  | $0.9573 \pm 0.12$   | $0.9742 \pm 0.08$   | $0.10 \pm 0.01$         | $0.18 \pm 0.04$              | $0.96 \pm 0.01$  | $0.89 \pm 0.07$        |
> | 3     | 50     | $0.13 \pm 0.02$               | $0.87 \pm 0.17$ | $0.85 \pm 0.19$  | $0.9318 \pm 0.01$   | $0.9619 \pm 0.01$   | $0.16 \pm 0.02$         | $0.22 \pm 0.06$              | $0.93 \pm 0.02$  | $0.99 \pm 0.14$           |
> | 3     | 80     | $0.15 \pm 0.02$               | $0.84 \pm 0.08$ | $0.83 \pm 0.10$  | $0.9223 \pm 0.07$   | $0.9550 \pm 0.09$   | $0.18 \pm 0.02$         | $0.25 \pm 0.13$              | $0.94 \pm 0.03$  | $1.07 \pm 0.25$           |
> | 3     | 100   | $0.17 \pm 0.02$               | $0.80 \pm 0.05$ | $0.81 \pm 0.02$  | $0.9131 \pm 0.02$   | $0.9565 \pm 0.02$   | $0.21 \pm 0.01$         | $0.29 \pm 0.10$              | $0.93 \pm 0.03$  | $1.25 \pm 0.19$           |
> | 3     | 200    | $0.16 \pm 0.07$               | $0.74 \pm 0.06$ | $0.72 \pm 0.04$  | $0.8950 \pm 0.02$   | $0.9603 \pm 0.03$   | $0.22 \pm 0.02$         | $0.35 \pm 0.12$              | $0.92 \pm 0.04$  | $1.45 \pm 0.16$           |
>
> > W2: Sparse or noisy time-series climate data may **hinder practical identifiability**.
>
> Thank you for raising this important point. We sincerely appreciate the insightful comment that learning **identifiable** representations from noisy and incomplete climate observations is a significant challenge in practice
> Indeed, we have made much effort to address these concerns. In particular:
>
> - For the **sparse** data, we do **not require invertibility** in the generative process, which offers potential resilience to partial observability or missing data.
> - For the **noisy** data, we allow the data-generating process to be **nonparametric** (Eq. (1)), which improves robustness to observational noise.
>
>
> In light of your insight, we will include related discussions about the identifiability under sparse/noisy observations in the revised manuscript.
>
> > Q1: How robust is CaDRe when some of the assumptions A1–A5 are violated, and can their violations be quantified or assessed in practice?
>
> Thanks for your insightful question! The assumptions A1–A5 primarily require **distributional variability** (e.g., nonstationarity and latent-driven dynamics), which are generally satisfied in real-world climate systems due to their inherent physical variability and forcing mechanisms, as discussed in **Main Paper, Line 141-146**.
>
> To quantify the degree of assumption violations and evaluate their impact, we conducted controlled simulation experiments where we violated A2, A3, A5, while noting that A1 (continuous density) and A4 (differentiability) are trivially satisfied in practice via neural network parameterizations, e.g., VAEs.
>
> The table below summarizes performance under these violations for $d_z = 3$ and $d_x = 6$, evaluating both representation quality and causal discovery:
>
> | Assumption                            | $d_z$ | $d_x$ | **SHD $(J_{\hat{g}}(\hat{x}_t))$** | TPR             | Precision       | MCC $(s_t)$        | MCC $(z_t)$        | SHD $(J_r(\hat{z}_t))$ | SHD $(J_d(\hat{z}_{t-1}))$ | $R^2$            |
> |:-------------------------------------:|:-----:|:-----:|:----------------------------------:|:---------------:|:----------------:|:------------------:|:------------------:|:----------------------:|:---------------------------:|:----------------:|
> | No Violation                          | 3     | 6     | $0.18 \pm 0.06$                    | $0.83 \pm 0.03$ | $0.80 \pm 0.04$  | $0.9583 \pm 0.02$   | $0.9505 \pm 0.02$   | $0.24 \pm 0.19$         | $0.33 \pm 0.09$              | $0.92 \pm 0.01$  |
> | Violate A2 (Contextual Variability)   | 3     | 6     | $0.26 \pm 0.07$                    | $0.71 \pm 0.05$ | $0.68 \pm 0.05$  | $0.7563 \pm 0.04$   | $0.8820 \pm 0.04$   | $0.36 \pm 0.07$         | $0.41 \pm 0.08$              | $0.67 \pm 0.03$  |
> | Violate A3 (Latent Drift)             | 3     | 6     | $0.31 \pm 0.05$                    | $0.67 \pm 0.13$ | $0.64 \pm 0.06$  | $0.8645 \pm 0.07$   | $0.8478 \pm 0.08$   | $0.39 \pm 0.12$         | $0.46 \pm 0.14$              | $0.78 \pm 0.21$  |
> | Violate A5 (Generation Variability)   | 3     | 6     | $0.35 \pm 0.11$                    | $0.65 \pm 0.12$ | $0.60 \pm 0.10$  | $0.7052 \pm 0.13$   | $0.9325 \pm 0.03$   | $0.41 \pm 0.08$         | $0.47 \pm 0.10$              | $0.85 \pm 0.02$  |
>
> The violations were implemented as follows:
> - **A2 (Contextual Variability)** was violated by using a non-injective latent transition $z_t = z_{t-1} + \varepsilon^z_t$ with uniform noise $\varepsilon^z_t$.
> - **A3 (Latent Drift)** was violated by modifying the generation to a nonlinear form $x_t = W z_t^2 + s_t$ with a linear matrix $W$.
> - **A5 (Generation Variability)** was violated by using a simplified linear additive form $s_t = z_t + \varepsilon^x_t$.
>
> These results show that CaDRe maintains meaningful performance under moderate violations, with consistently high $R^2$ and MCC scores. The magnitude of performance degradation provides a practical means to **quantify robustness**: violations of A2 and A3 mainly impair latent representation identification, while A5 primarily affects causal discovery.
>
> We have included these results and an extended discussion in the revised manuscript to clarify the empirical and theoretical robustness of our framework.
>
> > Q2: Physical interpretability of the latent variables and noise
>
> We appreciate this important question, which has given us the opportunity to deepen our analysis. We want to clarify that latent factors are, by definition, **unobserved**; in principle, we cannot give a direct interpretation. This poses a central challenge in the field of CRL. However, once we are sure about the existence of latent factors and understand how it's related to measured variables, we can begin to interpret them and even come up with ways to measure them. For example, as long as we obtain domain knowledge of latent factors from climate experts/scientists, we can easily match them to the meaningful quantities, e.g., precipitation, solar radiation, or components of a climate foundational model representation.
>
> Regarding the noise terms, they capture aggregated uncertainties from factors like human activity, measurement error, and unmodeled dynamics. While not directly interpretable, its distributional behavior (e.g., variability aligned with latent evolution) can still reveal useful scientific insights, such as how ocean currents influence the regional variability.
>
> These directions mirror historical processes in science, such as the discovery of viruses, which were first hypothesized based on indirect evidence and later confirmed and measured directly. We have included this perspective in our main paper and regard it as our future work.

---

> > ### Author Response · Authors · 2025-08-05
> > **Could you please let us know whether our responses properly addressed your concerns?**
> >
> > Dear Reviewer p8vt,
> >
> > We are grateful for your time on our paper, your constructive comments, and your recognition of the significance and novelty of our work. Could you please have a look at our response and let us know whether your concerns have been addressed, regarding
> >
> > - **W1:** Additional simulation results on higher-dimensional datasets to assess performance and efficiency scalability, showing that our model does not degrade in higher-dimensional settings with the physical prior.
> > - **W2:** Discussion of the robustness of our identifiability theory under sparse and noisy time-series climate data.
> > - **Q1:** Quantifying assumption violations through concrete cases, evaluating their impact on results via simulation experiments, which demonstrates the robustness of CaDRe.
> > - **Q2:** Clarification of the physical interpretability of latent variables and noise.
> >
> > Your further feedback would be highly appreciated.
> >
> > Best,
> >
> > The Authors of Submission 9138

---

> > ### Comment · Reviewer_p8vt · 2025-08-07
> >
> > Thanks for the authors' detailed rebuttal. I'll keep my score.

---

> > > ### Author Response · Authors · 2025-08-07
> > >
> > > Dear Reviewer p8vt,
> > >
> > > Thank you for taking the time to read our rebuttal. We sincerely appreciate your effort in reviewing our work.
> > >
> > > We would highly appreciate it if you could elaborate on the points that you are not satisfied with, as we can have the opportunity to provide further clarification and refine our work.
> > >
> > > With best regards,
> > >
> > > The Authors of Submission 9138

---

> ### Author Response · Authors · 2025-08-08
>
> Dear Reviewer p8vt,
>
> Thanks again for your time dedicated to reviewing this paper and for the strengths of the paper you formulated.  From our perspective, your comments are overall rather positive, with only two questions and two comments on the limitations.  As you see, we have provided detailed responses to them. We would highly appreciate it if you could have another look on our responses to confirm that all points have been addressed and consider updating your rating.
>
> Sincerely,
>
> The Authors of Submission 9138

---

### Official Review · Reviewer_2AQA · 2025-07-03

**Clarity:** 4
**Significance:** 3
**Originality:** 2
**Rating:** 5
**Confidence:** 2

**Summary:**

Causal discovery and representation learning are important approaches to better understand, for instance, Earth's climate or generally (spatio-)temporal data. The work builds a theory of when causal links are identifiable from such purely observational data while considering causal dependencies within and between both the observed and latent factors. The novel framework CaDRe is then composed of standard machine learning components and can uncover such causal graphs. Empirical findings on synthetic and real-world Earth data confirm the model's effectiveness.

**Questions:**

- Q1: What is the meaning of the arrow length in Fig. 6: The certainty of the edge existing in that orientation, or the strength of the influence?
- Q2: What are the latent factors that have been identified by CaDRe? The goal of CD/CRL in climate sciences is to uncover hidden structure, so having a latent representation we can understand is key.

**Ethical Concerns:**

["NO or VERY MINOR ethics concerns only"]

**Final Justification:**

The rebuttal was convincing. Some points were not fully resolved (e.g., interpretation of latent factors), but it is indeed material for future work.

I maintain my score and moderate confidence.

**Limitations:**

The discussion of limitations is very minimal. Aspects that are not sufficiently explored are robustness to noise and, generally, the set of assumptions necessary to run the method.

**Paper Formatting Concerns:**

**Minor Comments:**
- LL. 221 and 227: "Figure 4" links to Fig. 3.
- Table 3  is never referenced. It should be done in ll. 288ff.
- The table/figure ordering is slightly irritating and can be improved to align with the prose. For instance, Fig. 6 and Tab. 4 are discussed in the text in the opposite order to how they are presented. Which part of the figure shows latent transitions (cf. ll. 398f)?

**Quality:**

3

**Strengths And Weaknesses:**

**Strengths:**
- The paper is of high quality and written clearly.
- The problem being tackled is important. It is a significant contribution.
- The provided identification theory on latent variables is a substantial contribution, although in large parts deferred to the appendix. I did not check it in detail.
- The construction of the CD method is convincing and not more complex than necessary.
- The empirical findings support the methods' effectiveness.
- I am not sufficiently familiar with the field of CD/CRL to judge the overall novelty of the method nor the completeness of CD/CRL baselines.

**Opportunities for improvement (weaknesses):**
- The time-series forecasting baselines in Sec. 5.2 are not the latest (see, e.g., TimeMixer, xLSTM-Mixer, Chimera, or pretrained models such as Timer-XL). However, this does not substantially weaken the findings.
- While the key contributions are theoretical and on the modeling side (i.e., not the application), the core motivation for CaDRe was still climate science. However, only a single experiment on it is provided (CD in Sec. 5.2), which could be evaluated in more depth.
- The results on CD on actual climate data (Sec. 5.2) are solely qualitative. They lack contextualization of how existing models would fare. Additionally, a quantitative measure of correctness would be helpful as an aggregate indicator. This would be helpful since it is very hard to even qualitatively judge the appropriateness of the learned graph. Adding colors to denote correct/incorrect edges could be one more opportunity to improve this directly. The caption of Fig. 6/description in the text is overall rather minimal. See also Q1 below.
- No next steps (future work) are discussed.

---

> ### Author Rebuttal · Authors · 2025-07-31
>
> Dear Reviewer 2AQA, we sincerely appreciate your informative feedback that helps clarify our contributions and the completeness of our experiments. Here is the point-to-point response below.
>
> > W1 & W2: Lack of **latest** time-series forecasting baselines & Only a single experiment on climate science
>
> Thank you for raising these points, which help improve the soundness of our experiments. Please kindly note that we have considered **Weather** dataset is reported in **Appendix, Table A10**. In light of your suggestions, we further conduct experiments on CESM2, **Weather**, and an additional dataset **ERSST**, and reproduce the **Timer-XL**, **TimeMixer**, **TimeXer**, and **xLSTM-Mixer** for comparisons. Due to time limitations, we still work on Chimera due to the absence of released code, and will produce the experiment results as soon as possible. As shown in the table, CaDRe achieves competitive MSE and MAE across all datasets and forecast lengths.
>
> |Dataset|Length|CaDRe MSE|CaDRe MAE|TDRL MSE|TDRL MAE|CARD MSE|CARD MAE|FITS MSE|FITS MAE|MICN MSE|MICN MAE|iTransformer MSE|iTransformer MAE|TimesNet MSE|TimesNet MAE|Autoformer MSE|Autoformer MAE|Timer-XL MSE|Timer-XL MAE|TimeMixer MSE|TimeMixer MAE|TimeXer MSE|TimeXer MAE|xLSTM-Mixer MSE|xLSTM-Mixer MAE|
> |---|---|---|---|---|---|---|---|---|---|---|---|---|---|---|---|---|---|---|---|---|---|---|---|---|---|
> |CESM2|96|0.410|**0.483**|0.439|0.507|**0.409**|0.484|0.439|0.508|0.417|0.486|0.422|0.491|0.415|0.486|0.959|0.735|0.433|0.425|0.336|0.418|0.347|0.428|0.367|0.452|
> |CESM2|192|**0.412**|**0.487**|0.440|0.508|0.422|0.493|0.447|0.515|1.559|0.984|0.425|0.495|0.417|0.497|1.574|0.972|0.454|0.524|0.445|0.424|0.358|0.435|0.434|0.498|
> |CESM2|336|**0.413**|**0.485**|0.441|0.505|0.421|0.497|0.482|0.536|2.091|1.173|0.426|0.494|0.423|0.499|1.845|1.078|0.527|0.565|0.541|0.421|0.348|0.429|0.448|0.471|
> |Weather|96|**0.157**|**0.203**|0.442|0.511|0.423|0.497|0.172|0.221|0.199|0.256|0.168|0.214|0.180|0.231|0.225|0.259|0.367|0.252|0.367|0.252|0.367|0.252|0.367|0.252|
> |Weather|192|**0.207**|**0.248**|0.492|0.545|0.482|0.544|0.216|0.260|0.238|0.298|0.193|0.241|0.212|0.265|0.354|0.348|0.434|0.298|0.434|0.298|0.434|0.298|0.434|0.298|
> |Weather|336|**0.270**|**0.314**|0.536|0.612|0.525|0.596|0.386|0.439|0.316|0.496|0.426|0.494|0.423|0.499|0.354|0.348|0.527|0.565|0.341|0.421|0.348|0.429|0.375|0.341|
> |ERSST|96|**0.145**|0.268|0.187|0.268|0.197|0.273|0.539|0.297|0.726|0.765|0.247|0.264|0.432|0.508|0.953|0.272|0.163|**0.259**|0.172|0.272|0.365|0.344|0.345|0.255|
> |ERSST|192|**0.208**|0.307|0.214|**0.293**|0.233|0.375|0.226|0.752|1.263|0.892|0.251|0.535|0.452|0.585|1.024|0.908|0.210|0.294|0.214|0.302|0.372|0.367|**0.371**|0.297|
> |ERSST|336|**0.305**|0.361|0.462|0.388|0.487|0.484|0.439|0.535|1.173|1.172|0.305|0.659|0.581|0.607|1.387|1.353|0.352|**0.337**|0.439|0.394|0.429|0.448|0.476|0.357|
>
> The dataset statistics are as follows:
> - *Weather* dataset contains 52696 time steps starting from 2020-01-01 00:10:00 at regular 10-minute intervals. It includes 22 meteorological variables such as atmospheric pressure, temperature, humidity, vapor pressure, wind speed and direction, radiation, and photosynthetically active radiation (PAR). The data were collected from an automated rooftop station at the Max Planck Institute for Biogeochemistry in Jena, Germany.
>
> - *ERSST* dataset is from NOAA GlobalTemp (NOAA/NCEI) official website, we use the NOAA Global Temperature Anomaly Dataset (1880–2025), which includes 2052 monthly steps and 16,020 spatial grid points per step. For time-series forecasting, we use a downscaled version with 100 dimensions, obtained by averaging over block regions.
>
> [1] Wang, Yuxuan, et al. “Timexer: Empowering transformers for time series forecasting with exogenous variables.” NeurIPS 37 (2024): 469–498.
>
> > W3: CD results in Sec. 5.2 are only qualitative, lacks baselines and quantitative metrics; Fig. 6 is hard to interpret.
>
> Thank you for the suggestion. We have added **quantitative evaluations** using two new metrics based on wind direction priors: **Wind-SHD (WSHD)** and **Wind-TPR (WTPR)**. WSHD measures the normalized SHD between the estimated graph $B$ and a wind-induced reference graph $B_{\text{ref}}$, while WTPR computes the recall of $B$ with respect to $B_{\text{ref}}$.
>
> These metrics provide meaningful physical correctness scores, allowing baseline comparison. As shown below, **CaDRe outperforms all baselines**, achieving the best alignment with the wind-induced causal graph:
>
> | Metric | CaDRe | FCI   | CDNOD | PCMCI | LPCMCI |
> |--------|-------|-------|--------|--------|---------|
> | WSHD ↓ | 0.012 | 0.028 | 0.031  | 0.024 | 0.019   |
> | WTPR ↑ | 0.532 | 0.236 | 0.251  | 0.198 | 0.274   |
>
> In light of your suggestion, we also revised **Fig. 6** by color-coding edges (green: correct, red: incorrect) based on $B_{\text{ref}}$, and expanded both its **caption** and **description** to improve interpretability.
>
> > W4: No future work
>
> Thank you for pointing this out. Our future work will focus on two directions:
> - **More general causal structure**: We aim to extend our framework to support time-lagged causal relations in the observed space and sparse transition/generation processes, to better reveal how latent variables govern observations. This includes developing general identifiability guarantees and scalable estimation methods.
>
> - **Scalability with pretrained climate models**: We will integrate our framework with pretrained foundation models such as ClimaX [2] and GenCast [3] by introducing our flow-based module and structural constraints as plug-in fine-tuning components. This allows for refining latent representations post-training and uncovering the underlying causal structure in climate data.
>
> We have discussed these topics in detail and highlighted them in our revised manuscript.
>
> [2] Nguyen, Tung, et al. "Climax: A foundation model for weather and climate." arXiv preprint arXiv:2301.10343 (2023).
>
> [3] Price, Ilan, et al. "Gencast: Diffusion-based ensemble forecasting for medium-range weather." arXiv preprint arXiv:2312.15796 (2023).
>
> > Q1: The meaning of the arrow length in Fig. 6
>
> Thank you for raising this important question. In the visualized wind system (Top of Fig. 6), longer arrow length represents stronger wind speed, indicating the strength of the wind flow.
>
> In the estimated causal graph (Bottom of Fig. 6), if $a$ causes $b$, then we draw an arrow from the location of $a$ to the location of $b$ on the map. The length reflects the spatial distance, not the causal strength, with a longer length indicating a longer distance between two regions.
>
> We have clarified this distinction and illustrated the visualization procedure in the revised manuscript.
>
> > Q2: Physical meaning of latent factors
>
> Thank you for this important question. Latent factors are, by definition, **unobserved**, in principle, we cannot give a direct interpretation. This poses a central challenge in the field of CRL. However, once we are sure about the existence of latent factors and understand how it's related to measured variables, we can begin to interpret them and even come up with ways to measure them. For example, as long as we obtain domain knowledge of latent factors from climate experts/scientists, we can easily match them to the meaningful quantities, e.g., precipitation, solar radiation, or components of a climate foundational model representation. This mirrors historical processes in science, such as the discovery of viruses, which were first hypothesized based on indirect evidence and later confirmed and measured directly.
>
> We have included this perspective in our main paper and regard it as our future work.
>
> > Comment (1): Figure/table reference and the order of figures
>
> Thank you for your careful reading and helpful suggestions. We have corrected the figure and table references, and reordered the figures to align with the flow of the text in the revised manuscript.
>
> > Comment (2): Which part of the figure shows latent transitions (cf. ll. 398f)?
>
> The **middle part of the figure** illustrates the latent transitions, which are indicated by the **black lines** connecting latent states $\hat{z}_ {t-1}$ and $\hat{z}_ {t}$.
>
> In light of your comment, we have highlighted this part of the figure for improved clarity in our revised manuscript.

---

> > ### Comment · Reviewer_2AQA · 2025-08-04
> >
> > Thank you very much for the detailed response. It convinced me that I should maintain my score of "5: Accept".
> >
> > Deciphering latent factors is a substantial research objective on its own, agreed.

---

> > > ### Author Response · Authors · 2025-08-05
> > > **Thank you again for your support!**
> > >
> > > Dear Reviewer 2AQA,
> > >
> > > Thank you so much for your time dedicated to reviewing this paper, as well as for your discussion and insightful comments. Indeed, we fully agree with you that deciphering latent variables is a challenging task in climate science, typically requiring domain expertise. We are truly happy that our responses addressed your concerns and are grateful for your insightful review.
> > >
> > > With best wishes,
> > >
> > > The Authors of Submission 9138

---

### Comment · Area_Chair_QTCK · 2025-08-01
**The time to start author-reviewer discussions**

Dear all reviewers,

The author rebuttal period has now concluded, and authors' responses are
available for the papers you are reviewing. The Author-Reviewer Discussion
Period has started, and runs until August 6th AoE.

Your active participation during this phase is crucial for a fair and
comprehensive evaluation. Please take the time to:

- Carefully read the author responses and all other reviews.
- Engage in a constructive dialogue with the authors, clarifying points,
  addressing misunderstandings, and discussing any points of disagreement.
- Prioritize responses to questions specifically addressed to you by the authors.
- Post your initial responses as early as possible within this window to
  allow for meaningful back-and-forth discussion.

Your insights during this discussion phase are invaluable.
Thank you for your continued commitment to the NeurIPS review process.

Bests,
Your AC

---

### Author Response · Authors · 2025-08-08
**Thank you letter and general response**

Dear Reviewers and AC,

We sincerely thank you for your time, effort, and constructive feedback during the review and discussion phases. Your comments have been invaluable in improving the clarity, empirical validation, and theoretical framing of our work. We are grateful for the recognition of our contributions and for the suggestions that motivated new experiments, clarifications, and discussions in the revised manuscript.

While most concerns have been addressed accordingly, for clarity to the AC and reviewers, we provide below a concise summary of our general response; full details, tables, and derivations are available in our original point-by-point replies.

**I. Clarification on Theoretical Contributions**

- **Robust identifiability in CRL & CD**: Non-parametric identifiability theory for latent-variable CRL and CD in climate time-series, enabling recovery of latent representations and observational causal graphs without assuming invertibility, allowing sparse/noisy data (**W2, Q1 to p8vt**).
- **Why train an ICA instead of directly training a SEM**: In nonparametric SEM $X = f(X, E)$, noise $E$ cannot be separated from $X$, making direct SEM training ill-posed. Nonlinear ICA can recover the latent sources (noise) under identifiability guarantees, enabling principled causal graph estimation (Experiments & Discussions in **W4 to 1X1S**).
- **Assumptions Discussion**: Assumptions A1–A5 capture climate properties such as 1st-order (extendable) Markov structure and local variability in latent dynamics. Compared to prior CRL works, A2 & A3 are much weaker than the common invertibility requirement (Appendix A.8). We evaluated robustness under controlled violations of A2, A3, A5 to assess their impact. CaDRe maintained high $R^2$ and MCC scores, with degradation patterns depending on which assumption was violated (**Q1 to p8vt**, **L2 to NFts**).

---

**II. Extended Empirical Validation**

- **New datasets and baselines**: We have added *Weather* and *ERSST* datasets, and reproduced recent strong baselines — *Timer-XL*, *TimeMixer*, *TimeXer*, and *xLSTM-Mixer* for forecasting; *TCDF*, *IDOL*, and *TDRL* for causal discovery (**W1–W3 to 2AQA**, **W2 to 1X1S**).
- **Quantitative causal discovery metrics**: Introduced *Wind-SHD* and *Wind-TPR* to assess alignment with wind-induced reference graphs, showing CaDRe achieves the best scores across baselines (**W3 to 2AQA**).
- **Robustness tests**: Conducted evaluations on higher-order latent dynamics, controlled violations of assumptions (A2, A3, A5), and scaling to $d_x=\{20, 50, 80, 100, 200 \}$, demonstrating only mild performance degradation (**Q1 to p8vt**, **Q2 to G8Kr**).
- **Efficiency analysis**: Measured training time, memory usage, and inference latency, showing CaDRe is competitive or faster than baselines in both forecasting and causal discovery tasks (**W5 to 1X1S**).
- **Dimensionality degradation and physical priors**: In high-dimensional climate datasets, both simulated and real-world, we incorporate spatial distance–based physical priors into CaDRe. This reduces spurious dependencies, mitigates local minima, and preserves performance when scaling to large $d_x$ (**W1 to p8vt**, **Q3 to G8Kr**).

---

**III. Physical Interpretation & Visualization**

- **Physical interpretation of latent variables**: Latent variables are unobserved by definition; their interpretation requires domain knowledge, similar to historical scientific discoveries. Once their existence and relation to observed variables are established, they can be matched to meaningful physical quantities (e.g., precipitation, solar radiation) with expert input. Noise terms capture aggregated uncertainties such as human activity and measurement error (**Q2 to p8vt**, **W1 & Q1 to G8Kr**, **W3(3) to 1X1S**).
- **Visualization**: Visualizations distinguish wind-field arrows, where length indicates wind speed, from estimated causal graph edges, where length indicates spatial distance between connected regions. Latent transitions are explicitly highlighted to clarify their role in the dynamic process (**Q1 to 2AQA**, **W3(4) to 1X1S**).

---

**IV. Future Work**

We will:
- **Extend CaDRe to more general structures**: Support time-lagged observed causal structures and sparse generation processes, with corresponding new identifiability results (**W4 to 2AQA**).
- **Integration with pretrained climate models**: Incorporate CaDRe into pretrained climate foundation models (e.g., ClimaX, GenCast) as a fine-tuning module for post-training causal structure recovery (**W4 to 2AQA**).

---

**V. Closing**

With added datasets and baselines, new physical metrics, robustness and scalability tests, efficiency analysis, and clarifications on identifiability and assumptions, we believe all major concerns are addressed. We again thank the AC and reviewers for their constructive feedback and welcome any final questions.

---

### Note · Authors · 2025-08-12

Dear AC and Reviewers,

We express our sincere gratitude to all the reviewers for their valuable insights and to the AC for the time and effort dedicated to handling our paper!

We appreciate that most comments highlighted the **strengths** of our work: a unified CRL and CD framework for latent-variable climate; the non-trivial proof of equivalent ICA to SEM and the efficiency of the proposed CD method; the nonparametric identifiability without invertibility; empirical effectiveness with domain-consistent causal graphs; and overall clear theoretical framing, high-quality writing, and broad applicability.

As an abstraction of the details in our **[Thank you letter and general response]**, we provide here a summary of point-by-point responses to address the major concerns:

- To improve **high-dimensional scalability** (p8vt W1, G8Kr Q3), we conducted experiments with spatial-distance priors, showing mild degradation and strong robustness.
- For **assumption robustness** (p8vt Q1, NFts L2), we performed simulations violating A2/A3/A5 and assumption discussions.
- We clarify the **ICA over SEM** choice (p8vt Q2, 1X1S W4) by providing theoretical justification and empirical results.
- To enhance **notation and theory clarity** (NFts W4/Q1), we improved Def. 5, the prior estimation section, and Jacobian notation.
- For **efficiency** (1X1S W5), we added training time, memory usage, and latency comparisons, demonstrating CaDRe’s computational advantage.
- Finally, for **real-world climate experiments**, we clarified **latent interpretability** (p8vt Q2, 2AQA Q2, G8Kr W1/Q1, 1X1S W3), added the Weather and ERSST datasets for strengthening **real-world evaluation** (2AQA W2, 1X1S W3), included Timer-XL, TimeMixer, TimeXer, xLSTM-Mixer (forecasting) and TCDF, IDOL, TDRL (CD) as more **recent baselines** (2AQA W1, 1X1S W2), and introduced Wind-SHD and Wind-TPR metrics for **quantitative CD evaluation** (2AQA W3, NFts Q4).

All the points have been updated in the revised manuscript.

In the end, reviewers confirmed that **most of their concerns were addressed** or that our rebuttal **convinced** them; all the reviewers **did not raise any further questions** in their responses. Hence, we believe the revision addresses all major concerns and strengthens our contributions.

We again sincerely thank the AC and reviewers for their engagement, suggestions, and support.

Best regards,

The Authors of Submission 9138

---

### Decision · Program_Chairs · 2025-09-17

**Decision:**

Reject

**Comment:**

This paper presents CaDRe, a novel and ambitious framework that aims to
unify causal representation learning and causal discovery for time-series
data, with a strong motivation from climate analysis. The authors provide a
theoretical foundation for their approach, establishing identifiability
conditions for simultaneously learning a latent dynamic process and the
causal graph over observed variables in a nonparametric setting.

The reviewers recognised the importance of the problem and appreciated the
theoretical contributions and the clarity of the presentation. However,
significant concerns were raised during the initial review phase. The
primary weaknesses identified were the limited empirical validation, which
included outdated baselines, a single real-world dataset, and a lack of
quantitative metrics for the causal discovery claims. Furthermore,
reviewers questioned the practical scalability of the method and,
critically, the physical interpretability of the learned latent variables,
which may be particularly concerned by readers targeting on scientific
discovery. The AC also noticed a perceived disconnect between the
theoretical identifiability analysis and the general construction and
running of the learning algorithm. The empirical evaluation suffers from a
mismatch between the paper's core contribution—causal structure
learning—and the chosen baselines. The authors primarily compare against
models designed for time-series forecasting (with forecasting metrics in
comparison), while omitting a significant number of classical and
state-of-the-art methods for causal discovery. As a result, the performance
evaluation for the central task of causal graph inference is inadequate,
even if it has results in Fig.5, and the paper's claims in this area are
not sufficiently substantiated.

The authors provided an exceptionally thorough rebuttal, adding a large
volume of new experiments with more datasets, updated baselines, new
metrics, and scalability analysis. While this effort was commendable and
swayed several reviewers, it did not lead to a clear consensus. Moreover,
the crucial issue of latent variable interpretability remains largely
unaddressed and is deferred to future work, which could temper the paper's
claimed impact on climate science. One reviewer remained unconvinced by the
rebuttal, maintaining a negative score. This lack of consensus, combined
with the major revisions required to address the initial evaluation's
shortcomings, suggests the paper may not yet be ready for publication.
The AC acknowledges the promising and ambitious nature of the work
presented. However, after careful evaluation and in light of the highly
competitive submission environment and the unfortunately limited acceptance
ratio, it was collectively determined that this paper, regrettably, cannot
be accepted for this round. We highly encourage the authors to build upon
the promising foundation of this work and consider resubmission to a future
venue, addressing any specific feedback from the reviewers.